# PPT: Patch Order Do Matters In Time Series Pretext Task

**Jaeho Kim**[1*]   **Kwangryeol Park**[1*]   **Sukmin Yun**[2]   **Seulki Lee**[1†]

[1]Ulsan National Institute of Science and Technology (UNIST)

[2]Hanyang University ERICA

`{kjh3690,pkr7098,seulki.lee}@unist.ac.kr`
`sukminyun@hanyang.ac.kr`

## ABSTRACT

Recently, patch-based models have been widely discussed in time series analysis. However, existing pretext tasks for patch-based learning, such as masking, may not capture essential time and channel-wise patch interdependencies in time series data, presumed to result in subpar model performance. In this work, we introduce *Patch order-aware Pretext Task (PPT)*, a new self-supervised patch order learning pretext task for time series classification. PPT exploits the intrinsic sequential order information between patches in time and channel dimensions of time series data, where model training is aided by channel-wise patch permutations. The permutation disrupts patch order consistency across time and channel dimensions with controlled intensity to provide supervisory signals for learning time series order characteristics. To this end, we propose two patch order-aware learning methods: patch order consistency learning, which quantifies patch order correctness, and contrastive learning, which distinguishes weakly permuted patch sequences from strongly permuted ones. With patch order learning, we observe enhanced model performance, e.g., improving up to 7% accuracy for the supervised cardiogram task and outperforming mask-based learning by 5% in the self-supervised human activity recognition task. We also propose ACF-CoS, an evaluation metric that measures the *importance of orderness* for time series datasets, which enables pre-examination of the efficacy of PPT in model training.

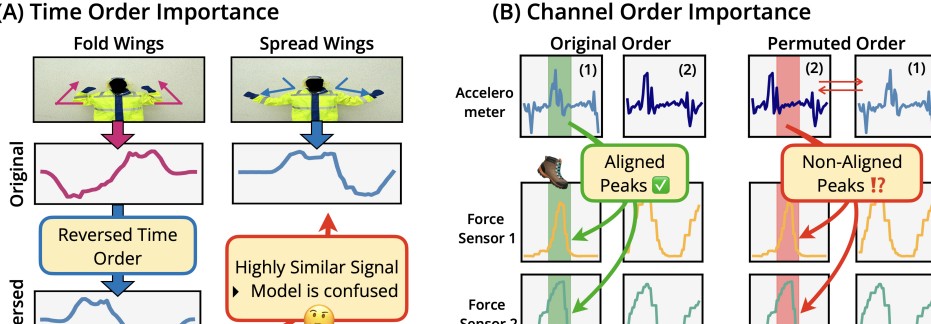

**Figure 1: Motivation.** Why is patch order important in time series? (A) **Time Order Importance.** The aircraft handling signal task (Song et al., 2011) shows how actions can be misinterpreted if their time order is ignored. For instance, "Fold Wings" and "Spread Wings" are distinct actions, but their time signals may appear similar when reversed, potentially causing confusion in model training and highlighting the need for time order awareness in classification. (B) **Channel Order (Alignment) Importance.** The smart shoe task (Kim et al., 2023) shows how permuting patch orders in one channel (accelerometer) can disrupt overall channel alignment. Peaks that were initially aligned across all channels during a stepping action (Green shaded) shift out of alignment (Red shaded) when the top two patches from the accelerometer are permuted, misaligning signals. Our proposed method, PPT, supervises this information during model training to address these challenges.

---

*Equal Contribution.

†Corresponding author

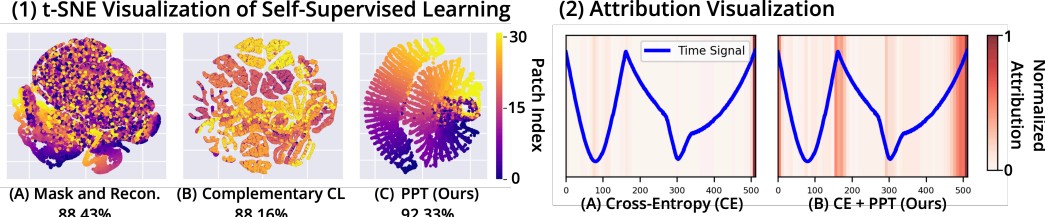

**Figure 2:** (1) t-SNE (Van der Maaten & Hinton, 2008) visualization of patches from the smart insole task (Kim et al., 2023). All figures are derived from self-supervised linear probing using PatchTST (Nie et al., 2022), color-coded by patch index. (A) uses a mask-and-reconstruct pretext task (Nie et al., 2022). (B) implements the Complementary patch contrastive learning (CL) proposed by (Lee et al., 2023). (C) uses PPT (Patch order aware Pretext Task), where we observe a clear alignment of patch orders in latent space. (2) Feature ablation (Kokhlikyan et al., 2020) to find patch importance in supervised training. Results on the ShapesAll task (Dau et al., 2019) reveals high importance scores at key time segments when using PPT compared to using cross-entropy (CE) alone. The details of the results are shown in Appendix A.

# 1 INTRODUCTION

Recent advances in time series have seen a notable trend toward the adoption of patch-based methodologies (Cao et al., 2023; Chen et al., 2023; Lee et al., 2023; Wang et al., 2023). These methods segment the time series into "patches," each consisting of consecutive data points, moving away from the traditional "temporal-wise" or "point-wise" approaches that model individual time points separately. Although the shift to using patches has generally improved model performance, current approaches used in patch-based models often struggle to effectively incorporate order information between time series patches. Positional encoding (Vaswani et al., 2017) and strategies such as mask-and-reconstruct (Nie et al., 2022) or patch complementary contrastive learning (CL) (Lee et al., 2023) still fail to fully capture the chronological and channel-specific order alignment essential in time series (Fig. 2-(1)). Modeling this order information is important as it leverages the unique characteristics of time series, specifically, their inherent sequential nature. By improving order awareness across patches, a model can improve its performance, ensuring that the temporal dynamics and channel dependencies characteristic of time series are effectively captured and utilized.

To address this gap, we propose Patch order-aware Pretext Task (PPT), the first order learning pretext task for patch-based time series model, which creates supervising signals **from the inherent order relationship between the time series patches**. This order awareness across time and channel is achieved through channel-wise permutation of patches, where the "channel" refers to a specific feature (e.g., sensor) in time series. The motivation of PPT comes from a critical observation: **The sequence and arrangement of time series patches, spanning both the time and channel dimensions, represent a natural inductive bias that can be exploited in learning**. This awareness is crucial, as time series data exhibit complex interdependencies that are necessary for accurate classifications. For instance, as shown in (Fig. 2-(2)), visualization of the attribution scores for each patch demonstrates that incorporating PPT effectively identifies inflection points, while using cross-entropy (CE) as a standalone loss function fails to highlight meaningful time segments. Failure to account for these dependencies can lead to suboptimal model performance, diminishing the model's ability to understand the underlying data structure, and reducing accuracy and reliability in tasks where order is crucial. We further illustrate our motivations with real-world examples in Fig. 1.

A pretext task is a *"pre-designed tasks for a network to solve, where the supervising signal is generated from the data itself"* (Jing & Tian, 2020). As a pretext task, PPT is adaptable to self-supervised learning (SSL) and supervised learning. In SSL, PPT operates independently as a standalone loss function, while in supervised training, it supplements the standard cross-entropy (CE) minimization as an additional learning task without relying on labeled information. PPT is a plug-in method for patch-based time series models that encode individual patches into latent representations (e.g., PatchTST, PITS). By integrating PPT, the model gains insight into the relationships of structural order between patches, encompassing both the time and channel dimensions, resulting in consistent improvements in model performance. Furthermore, we introduce ACF-CoS (Autocorrelation Function with Cosine Similarity), a metric based on autocorrelation function, to quantify the "orderness" of a time series. Briefly, orderness is the ability of a discriminative function to distinguish a sequence from its permuted

---

[0] https://github.com/eai-lab/PPT

set, emphasizing sequence order and structural dependency (Definition 3.1). An extensive number of experiments demonstrate a positive correlation between the ACF-CoS score and the performance enhancement brought by PPT, underlining its significance in exploiting structural data characteristics for patch-based time series. The key contributions of our work are summarized as follows.

- We introduce Patch order-aware Pretext Task (PPT), the first order-aware pretext task for patch-based time series models. As a self-supervised learning method, we apply and assess PPT on two state-of-the-art patch-based time series models, i.e., Transformer-based PatchTST (Nie et al., 2022) and linear-based PITS (Lee et al., 2023), demonstrating PPT's effectiveness.
- We propose two patch order-aware learning methods of PPT, i.e., patch order consistency and contrastive learning, which exploits inherent structural orders between patch sequences across time and channel dimensions, applicable to both self-supervised and supervised training.
- We propose ACF-CoS (Autocorrelation Function with Cosine Similarity), a metric for gauging the "orderness" of a time series. It enables us to pre-examine whether a time series task could benefit from patch order learning, enabling targeted applications of PPT where the order is critical.

## 2 RELATED WORKS

**Self-Supervised Learning in Time Series.** TS-TCC (Eldele et al., 2021) is a self-supervised learning (SSL) method designed for non-patch-based time series, generating two views of the time series data using weak (jittering and scaling) and strong (permuting and jittering) augmentations. Then, an autoregressive model is used to construct a context vector for temporal contrast between the two samples. Based on that, some semi-supervised (i.e. CA-TCC (Eldele et al., 2023)) and graph-structured learning (i.e. TS-GAC (Wang et al., 2023)) methods have followed. Inspired by TS-TCC, we propose PPT that introduces a channel-wise patch permutation approach with varying strengths determined by the permutation frequency, termed 'weak' and 'strong' permutation. While the main objective of TS-TCC is to bring the weak and strong representations closer, the objective of PPT is the opposite, as it widens the distance between the weak and strong while placing the weak closer to the non-permuted 'original' set (Sec. 3.2). Moreover, unlike TS-TCC's random segment permutations (Um et al., 2017) made on the fly, PPT provides control over the strength of permutation, which enables the proposed time and channel-wise consistency and contrastive learning (Sec. 3.3). In addition, PPT is not only efficient due to its use of the permutation bank (i.e. only the permuted indices are stored) but also effective because it continuously exposes the model to a diverse array of permuted samples. Thus, PPT significantly diverges from TS-TCC in its focus for contrasting, making it a unique and effective patch-based pretext task for time series.

**Patch-Based Representation of Time Series.** Patch-based methods have gained significant popularity across various time series applications, including time-series foundation models (Jin et al., 2023; Zhou et al., 2024; Gao et al., 2024), forecasting (Chen et al., 2023), and classification (Cheng et al., 2023). TimeMAE (Cheng et al., 2023) have integrated the Masked Autoencoder (He et al., 2022) architecture for time series, optimizing the reconstruction loss of randomly masked patches. PatchTST (Nie et al., 2022), a Transformer-based model, utilizes mask reconstruction as a self-supervised pretraining method, establishing itself as a strong baseline for various time series applications. TS-GAC (Wang et al., 2023) is a graph-based model where patches are represented as nodes and their correlations as edges, performing contrastive learning in both node and graph levels. PITS (Lee et al., 2023), an MLP-based model, enhances self-supervised training by combining contrastive learning and masked modeling of patches. As a simple yet effective self-supervised learning method, PPT is applicable to many of those patch-based time series models.

**Quantifying Time Series.** Time series exhibit substantial variations in length, number of channels, and dynamics. As such, quantifying these series into a single metric provides an interpretable tool for understanding both the data and the behavior of models applied to it. Shannon entropy (SE) (Lin, 1991) and its temporal extension, Permutation entropy (PE) (Bandt & Pompe, 2002), are foundational metrics used to measure the uncertainty and pattern frequency in time series. However, SE and PE may not adequately capture the intrinsic orderness of data sequences. Addressing this gap, we introduce the ACF-CoS metric (Sec. 4), which quantifies the orderness of a time series by computing the cosine similarity between autocorrelation functions of the original and its permuted (shuffled) set.

**Extended Related Works.** Appendix B discusses related works for order-aware learning tasks in domains other than time series and SSL methodologies for time series in general.

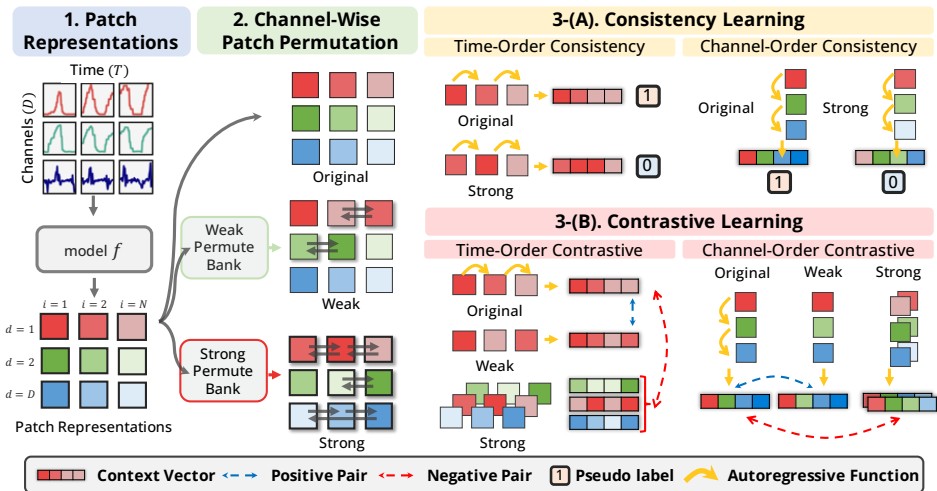

**Figure 3:** PPT employs three primary components: (1) **Patch Representations.**, time series patches are encoded into latent patch representations; (2) **Channel-Wise Patch Permutation.**, permuting patch orders channel-wise with permutation intensity governed by weak or strong criteria; (3-A) **Consistency Learning.**, an autoregressive approach for recognizing patch order at both time and channel levels, paired with (3-B) **Contrastive Learning.** to distinguish between original/weakly permuted (positive) and strongly permuted (negative) patches of sequences. For the negatives in contrastive learning, we use strongly permuted sequences from all channels. Best in color.

# 3  PPT: Patch Order-aware Pretext Task for Time Series Classification

**Preliminaries.** Let $\mathbf{X} \in \mathbb{R}^{T \times D}$ be a raw time series, where $T$ and $D$ denote the time sequence length and the number of channels (features), respectively. For each channel, we derive $N = \lfloor \frac{T}{s} \rfloor$ non-overlapping patches, each of size $s$. The latent representation for the $i$-th patch from the $d$-th channel in $\mathbf{X}$ is denoted by $\mathbf{p}_{(i,d)} \in \mathbb{R}^{d_h}$, where $d_h$ is the dimensionality of the embedding, obtained via a model $f(\cdot)$. The sequence of patches in time order can be obtained for each channel $d \in [1:D]$ and is represented as $\mathbf{P}_{(:,d)} = \langle \mathbf{p}_{(1,d)}, \dots, \mathbf{p}_{(N,d)} \rangle \in \mathbb{R}^{d_h \times N}$. Additionally, a sequence of patches in channel order can be obtained for all $i \in [1:N]$ and is represented as $\mathbf{P}_{(i,:)} = \langle \mathbf{p}_{(i,1)}, \dots, \mathbf{p}_{(i,D)} \rangle \in \mathbb{R}^{d_h \times D}$. While time order $\mathbf{P}_{(:,d)}$ captures the temporal evolution within individual channels, the channel order sequence $\mathbf{P}_{(i,:)}$ represents a snapshot of patterns that appear together across all channels at a single time point, enabling the model to learn both temporal dynamics and concurrent cross-channel relationships. The objective of PPT is to exploit these order relationships in the time and channel dimensions of the patches derived solely from $\mathbf{X}$. Here, we define patch order as:

**Definition 3.1 (Order)** *Let $\mathcal{P}_{time}$ and $\mathcal{P}_{channel}$ be domains of time and channel order patch sequences, respectively. We define a discriminative function $g(\cdot)$ such that $g : \mathcal{P} \to \mathbb{R}$ and permute operation $\Pi : \mathcal{P} \to \mathcal{P}$, where $\mathcal{P} = \mathcal{P}_{time} \cup \mathcal{P}_{channel}$. An order exists in $\mathbf{P}$, where $\mathbf{P}$ is an element of $\mathcal{P}$, if the discriminative function $g$[1] can distinguish between $\mathbf{P}$ and its permuted $\Pi(\mathbf{P})$.*

**Remark 3.1** *The discriminatory power of function $g$ suggests a non-random order or sequence dependency in $\mathbf{P}$, which is disrupted through $\Pi(\mathbf{P})$.*

## 3.1  Structure Overview

We provide an overview of PPT in Fig. 3, outlining a three-step process. (1) Time series are reordered into sequences of patches, with each patch transformed into patch embeddings through encoder $f$. (2) utilizing a pre-built permutation bank, patches undergo efficient channel-wise permutations within each channel, resulting in two permuted orders of patch sequences-termed "Weak" and "Strong" based on permutation intensity (Sec. 3.2). (3) These sequences are then applied in consistency and contrastive learning (Sec. 3.3), where consistency learning verifies the correctness of the order of the patch sequence, and contrastive learning pairs the Original and Weak sequences as positives against the Strong sequence as negative. The learning occurs in both the time and channel dimensions.

---

[1]In this paper, $g$ refers to an LSTM module for PPT training, and to the ACF-CoS in ACF-CoS calculation.

## 3.2 CHANNEL-WISE PATCH PERMUTATION STRATEGY

**Permutation Strategy.** We propose a straightforward yet effective patch permutation strategy wherein patches are permuted within each channel. **Permuting within each and every channel** is sufficient to break the patch consistency in both time and channel dimensions. The permutation operator $\Pi_\gamma \left( \mathbf{p}_{(i,d)}, \mathbf{p}_{(j \neq i,d)} \right)$ selects and permutes the order of two distinct patches, $\mathbf{p}_{(i,d)}$ and $\mathbf{p}_{(j \neq i,d)}$, within the same channel $d$. The hyperparameter $\gamma$ regulates the frequency of permutations performed for each channel. For instance, $\gamma = 1$ means that a single pair of patches within each channel will be permuted. We propose and investigate three selection methods that determine which patch pairs to be permuted, i.e., the vicinity, distant, and random pairs (See Appendix C). Our experiment shows that the random pair selection is most effective, as it encompasses both the vicinity and distance pairs. Consequently, we employ random pair selection for our patch permutations.

Furthermore, we define the permuted **sequence of patches** as "Weak" and "Strong" based on the magnitude of $\gamma$. Selecting a small $\gamma$ leads to the permutation of only a few pairs of patches within a channel, maintaining most of the original patch sequence order. In contrast, a high value of $\gamma$ substantially disrupts the order of the original sequence in both the time and channel. We utilize "Weak," "Strong," and the un-permuted "Original" sequence of patches in the training of PPT.

**Permutation Time Complexity.** The permutation operation is efficiently managed using a pre-built bank of permuted indices, with each index randomly sampled to maintain diversity during minibatch training. In theory, permuting patch positions with respect to the number of patches $N$ only involves reordering them, which has a time complexity of $\mathcal{O}(N)$ for each permutation. In practice, permuting a batch of 512 data samples takes merely 0.018 seconds on modern GPU hardware (Appendix F). The time complexity and the pseudo-algorithm for the permutation bank are in Appendix D.

## 3.3 CONSISTENCY AND CONTRASTIVE LEARNING

**Time and Channel-Order Patch Consistency.** PPT learns to differentiate between the "Original" and "Strong" patch sequences in both time and channel dimensions. It is referred to as learning "consistency" in the order of patches. The motivation for this learning objective is based on the observations (see Fig. 1), where an intrinsic order exists among patches, and any deviation from this order may result in patch sequences that appear to be inconsistent and/or errant. To quantify the consistency, autoregressive models $g_{\texttt{Time}}$ and $g_{\texttt{Channel}}$ summarize the patch sequences $\mathbf{P}_{(i,:)i \in [1:N]}$ and $\mathbf{P}_{(:,d)d \in [1:D]}$ into context vectors $\mathbf{c}_{i \in [1:N]} \in \mathbb{R}^{d_h}$ and $\mathbf{c}_{d \in [1:D]} \in \mathbb{R}^{d_h}$, respectively. Here, we employ single-layer uni-directional LSTMs that process the patch sequences. The final cell state of each LSTM serves as the context vector, effectively summarizing the patch sequences. These LSTM modules are trained end-to-end with the backbone model (details in Appendix F). The context vectors are then utilized in pseudo-label training, where the learning task is to discriminate between the context vectors of the "Original (label 1)" and "Strongly (label 0)" permuted sequences. Based on the binary cross entropy, we define the consistency loss for time and channel-order patches as:

$$\mathcal{L}^{\text{CS}}_{\texttt{Time}} \text{ or } \mathcal{L}^{\text{CS}}_{\texttt{Channel}} = -\frac{1}{m} \sum_{i=1}^{m} \left[ y_i \log(\hat{y}_i) + (1 - y_i) \log(1 - \hat{y}_i) \right]. \tag{1}$$

where $\mathcal{L}^{\text{CS}}_{\texttt{Time}}$ with $\mathbf{c}_{i \in [1:N]}$ and $\mathcal{L}^{\text{CS}}_{\texttt{Channel}}$ with $\mathbf{c}_{d \in [1:D]}$, $m$ is the number of patch sequence samples either in time and channel, $y$ and $\hat{y}$ are the pseudo-label and predicted label, respectively. The total consistency loss $\mathcal{L}^{\text{CS}}_{\text{sum}}$ is then the sum of the losses from both dimensions, $\mathcal{L}^{\text{CS}}_{\text{sum}} = \mathcal{L}^{\text{CS}}_{\texttt{Time}} + \mathcal{L}^{\text{CS}}_{\texttt{Channel}}$.

**Time and Channel-Order Patch Contrasting.** The use of the consistency loss, $\mathcal{L}^{\text{CS}}$, empowers the model to discern between the original and permuted sequences of patches in both time and channel dimensions. To further refine the model's ability to detect varying degrees of permutation, we introduce an InfoNCE loss (Oord et al., 2018), denoted as $\mathcal{L}^{\text{CT}}$, to detect the degree of permutation variations. Here, the "Original" serves as the anchor, the "Weak" as positives, and the "Strong" as negatives. Autoregressive models are employed for contrastive learning, taking context vectors to distinguish between degrees of permutation. We define the contrastive loss for time order $\mathcal{L}^{\text{CT}}_{\texttt{Time}}$ in Eq. (2), where $D$ is the number of channels, $\mathbf{c}$ is the context vectors, $\tau$ represents the temperature hyperparameter, and sim denotes cosine similarity. Similarly, the contrastive loss for channel order $\mathcal{L}^{\text{CT}}_{\texttt{Channel}}$ is defined as Eq. (2) by replacing $D$ with the number of patches $N$. Then, the total contrastive loss becomes the sum of the two losses as $\mathcal{L}^{\text{CT}}_{\text{sum}} = \mathcal{L}^{\text{CT}}_{\texttt{Time}} + \mathcal{L}^{\text{CT}}_{\texttt{Channel}}$, measured in both

**A) Patch Embeddings in Time-Level**

PatchTST : 78.6%    PatchTST + PPT: 81.5% (+ 2.9%)

● Index: 1 - 4    ● Index: 13 - 16    ● Index: 27 - 30

**B) Patch Embeddings in Channel-Level**

PatchTST: 91.6%    PatchTST + PPT: 97.4% (+ 5.8%)

● Accelerometer Sensors    ● Force Sensors

**Figure 4:** t-SNE (Van der Maaten & Hinton, 2008) visualization of patch embeddings from supervised PatchTST (Nie et al., 2022). **(A) Time-Level Patch** from the ECG data (Bousseljot et al., 1995) visualize temporal patch embeddings with patch index from $i = 1 - 4$ (Green), $13 - 16$ (Yellow), and $27 - 30$ (Purple). **(B) Channel-Level Patch** from the smart insole task (Kim et al., 2023) shows the distinct embeddings for the accelerometer channels (Blue) and force sensor channels (Red). We observe a clear distinction between time and channel patches with the use of PPT, with performance increases detailed in Tab. 3.

time and channel dimensions.

$$\mathcal{L}_{\text{Time}}^{\text{CT}} = -\frac{1}{D}\sum_{d=1}^{D}\log\left(\frac{\exp(\text{sim}(\mathbf{c}_d^{\text{original}}, \mathbf{c}_d^{\text{weak}})/\tau)}{\exp(\text{sim}(\mathbf{c}_d^{\text{original}}, \mathbf{c}_d^{\text{weak}})/\tau) + \sum_{k=1}^{D}\exp(\text{sim}(\mathbf{c}_d^{\text{original}}, \mathbf{c}_k^{\text{strong}})/\tau)}\right) \quad (2)$$

Although consistency and contrastive learning losses are presented here for a multivariate time series, they can be applied to a univariate time series by removing channel losses $\mathcal{L}_{\text{Channel}}^{\text{CS}}$ and $\mathcal{L}_{\text{Channel}}^{\text{CT}}$.

### 3.4 OVERALL LOSS SETUP

**Self-Supervised Loss.** In self-supervised (SSL) training, we focus on optimizing the consistency and contrastive terms in Eq. (3). Here, $\lambda_1$ and $\lambda_2$ are hyperparameters. However, to avoid extensive manual tuning, they are replaced by learnable parameters that dynamically adjust the weight of each loss term, reflecting a strategy based on homoscedastic uncertainty for multi-task learning (Kendall et al., 2018; Dong et al., 2024).

**Supervised Loss.** For supervised training, PPT works as an extra loss term along with the task-specific supervised loss $\mathcal{L}_{\text{T}}$, the cross-entropy loss in the case of classification. For instance, Fig. 4 compares the t-SNE embeddings of patches with and without PPT in a supervised setup. The final objective function is Eq. (3), with the hyperparameter $\lambda$ replaced by learnable parameters as in SSL.

$$\mathcal{L}_{\text{Self-Supervised}} = \lambda_1\mathcal{L}_{\text{sum}}^{\text{CS}} + \lambda_2\mathcal{L}_{\text{sum}}^{\text{CT}}, \text{ and } \mathcal{L}_{\text{Supervised}} = \mathcal{L}_{\text{T}} + \lambda_1\mathcal{L}_{\text{sum}}^{\text{CS}} + \lambda_2\mathcal{L}_{\text{sum}}^{\text{CT}} \quad (3)$$

## 4 ACF-COS METRIC: MEASURING THE IMPORTANCE OF ORDERNESS

We propose ACF-COS (Autocorrelation Function with Cosine Similarity) to measure the importance of orderness in time series data. This score allows us to pre-assess the suitability of PPT as a pretext task for a given dataset. For a time series $\mathbf{x} = [x_1, \ldots, x_T]^\top$ given the mean $\bar{x}$, the $j$-th element $a_j$ of the autocorrelation vector $\mathbf{a} = [a_0, \ldots, a_{T-1}]^\top$ is defined as in Eq. (4), with $j$ as lag. We calculate two distinct autocorrelation vectors, $\mathbf{a}$ and $\mathbf{a}'$, from the original time series $\mathbf{x}$ and its patch-permuted counterpart $\mathbf{x}'$, respectively. While the calculation of $\mathbf{a}$ from the original time series $\mathbf{x}$ is straightforward, $\mathbf{a}'$ is obtained from the patch-permuted series $\mathbf{x}'$ by (1) reshaping $\mathbf{x}$ into a sequence of patches, (2) randomly shuffling those patches, and (3) reassembling them into the final series $\mathbf{x}'$. The ACF-COS score is then computed by taking the cosine similarity between $\mathbf{a}$ and $\mathbf{a}'$ as:

$$\text{ACF-COS}(\mathbf{a}, \mathbf{a}') = 1 - \mathbf{a} \cdot \mathbf{a}'/\|\mathbf{a}\|\|\mathbf{a}'\|, \; a_j = \sum_{t=1}^{T-j}(x_t - \bar{x})(x_{t+j} - \bar{x})/\sum_{t=1}^{T}(x_t - \bar{x})^2. \quad (4)$$

The interpretation of ACF-COS is intuitive and aligned with the definition of patch order (Definition 3.1). A low ACF-COS score indicates that the two autocorrelation vectors are similar, meaning that permuting the sequences does not significantly alter their autocorrelation, suggesting a weak structural order dependency. In contrast, a high ACF-COS shows that permutation effectively disrupts autocorrelation, indicating a strong dependence on the sequence order. Consequently, white noise has an ACF-COS score close to zero, which means that there is no structural order dependency, whereas the step function shows ACF-COS close to one, which implies a significant dependency. To quantify the order characteristics of the data, we treat each channel as an independent sequence and average the ACF-COS scores of all data samples to obtain the final ACF-COS score (Tab. 4). To mitigate the impact of patch size variations, we calculate the ACF-COS scores over different patch sizes.

**Table 1: Self-Supervised Linear Probing Results.** We compare PPT (ours) to state-of-the-art self-supervised learning methods. The asterisk (*) denotes patch-based methods. The baselines PatchTST* and PITS* use self-supervised training methods from their original work. The best scores are highlighted with **red bold**. The full experimental result, including the MS-HAR task, is given in Appendix J.

| Dataset | Models | Accuracy | F1 score | AUROC | AUPRC | Precision | Recall |
|---|---|---|---|---|---|---|---|
| EMO | Mixing-up | 74.48±2.93 | 49.30±2.53 | 71.80±7.11 | 52.22±2.71 | 51.66±1.81 | 49.36±4.01 |
| | SimCLR | 74.42±4.38 | 44.64±6.34 | 72.71±7.29 | 48.34±6.76 | 47.83±7.07 | 47.19±8.02 |
| | TS2Vec | 78.08±2.93 | 49.07±3.00 | 78.00±3.58 | 50.95±3.83 | 48.97±3.52 | 49.87±2.88 |
| | TF-C | 77.63±6.18 | 53.30±6.90 | 82.22±5.31 | 54.84±7.18 | 56.14±8.79 | **56.00±9.07** |
| | TS-TCC | 75.61±3.49 | 48.47±3.59 | 73.82±7.40 | 53.98±3.77 | 54.75±0.67 | 49.02±4.05 |
| | SimMTM | 81.75±3.33 | 53.08±3.68 | 81.35±7.69 | 58.70±4.03 | 57.75±4.84 | 51.62±4.17 |
| | TimeMAE* | 73.97±2.28 | 42.44±2.07 | 70.11±4.43 | 43.25±2.14 | 42.81±2.37 | 42.43±2.09 |
| | TS-GAC* | 73.75±1.66 | 46.42±1.29 | 75.92±2.30 | 49.29±0.62 | 46.04±0.87 | 48.86±1.69 |
| | UniTS* | 78.37±0.13 | 29.51±0.51 | 65.95±2.95 | 44.84±2.39 | 29.47±7.52 | 33.44±0.24 |
| | PatchTST* | 78.70±0.73 | 45.81±2.07 | 82.60±1.39 | 55.23±2.21 | 59.40±5.32 | 46.35±1.31 |
| | PatchTST (+PPT)* | **81.92±0.58** | **54.19±2.33** | **84.74±1.55** | **62.51±3.09** | **62.96±2.49** | 53.41±2.42 |
| | PITS* | 69.63±2.04 | 43.73±1.06 | 68.84±2.39 | 43.90±1.05 | 44.07±0.82 | 45.68±1.31 |
| | PITS (+PPT)* | 75.55±2.84 | 45.75±2.43 | 68.63±3.30 | 45.59±2.28 | 45.05±2.65 | 47.53±2.06 |
| PTB | Mixing-up | 82.35±0.93 | 88.82±0.57 | 89.40±2.22 | 95.32±1.05 | 81.76±0.72 | 97.23±0.61 |
| | SimCLR | 80.41±1.14 | 87.89±0.62 | 86.03±3.24 | 93.84±1.71 | 79.30±0.99 | 98.57±0.21 |
| | TS2Vec | 84.06±2.81 | 89.75±1.68 | 91.56±3.28 | 96.17±2.02 | 83.82±2.56 | 96.62±0.95 |
| | TF-C | 79.58±1.22 | 87.25±0.85 | 86.55±2.64 | 93.95±1.50 | 79.35±0.47 | 96.91±1.78 |
| | TS-TCC | 79.58±1.18 | 86.98±0.78 | 79.18±1.88 | 88.05±1.40 | 80.51±0.90 | 94.59±1.47 |
| | SimMTM | 80.75±0.29 | 88.08±0.17 | 88.66±1.96 | 94.48±1.23 | 79.58±0.31 | 98.61±0.41 |
| | TimeMAE* | 78.65±1.43 | 86.78±0.86 | 84.72±1.30 | 92.97±0.83 | 78.41±1.03 | 97.15±1.10 |
| | TS-GAC* | 83.26±1.97 | 89.51±1.09 | 91.18±2.36 | 95.37±0.96 | 81.76±1.95 | **98.90±0.23** |
| | UniTS* | 84.20±1.01 | 89.57±0.60 | 88.98±1.35 | 94.81±0.84 | 85.51±1.08 | 94.04±0.23 |
| | PatchTST* | 79.89±1.87 | 87.46±1.07 | 81.36±4.91 | 88.94±3.50 | 79.50±1.54 | 97.21±1.07 |
| | PatchTST (+PPT)* | 78.50±1.76 | 88.67±0.95 | 76.04±6.47 | 86.79±5.56 | 78.45±1.62 | 96.83±0.63 |
| | PITS* | 84.61±1.21 | 89.96±0.61 | 86.53±2.48 | 92.75±1.58 | 85.03±2.00 | 95.55±1.32 |
| | PITS (+PPT)* | **86.48±0.40** | **91.24±0.26** | **91.83±1.36** | **96.26±0.85** | **85.67±0.61** | 97.58±0.86 |
| GL | Mixing-up | 81.76±2.94 | 79.25±3.85 | 96.46±0.50 | 82.43±2.80 | 79.99±3.83 | 79.18±3.93 |
| | SimCLR | 87.10±0.44 | 88.72±0.79 | 98.25±0.12 | 92.69±1.04 | 89.42±0.84 | 88.20±0.84 |
| | TS2Vec | 87.79±0.66 | 89.92±0.68 | 98.35±0.12 | 93.93±0.10 | 90.12±0.97 | 89.86±0.42 |
| | TF-C | 88.23±0.73 | 89.09±1.11 | 98.66±0.13 | 92.90±1.00 | 89.30±1.12 | 89.00±1.02 |
| | TS-TCC | 87.98±1.28 | 88.53±1.32 | 98.60±0.22 | 91.86±1.03 | 88.82±1.45 | 88.36±1.27 |
| | SimMTM | 82.05±0.33 | 83.96±0.50 | 97.08±0.06 | 88.06±0.49 | 84.63±0.64 | 83.63±0.47 |
| | TimeMAE* | 83.44±1.09 | 85.26±1.74 | 97.81±0.11 | 89.84±1.09 | 86.10±2.06 | 84.77±1.58 |
| | TS-GAC* | 92.32±0.85 | 92.79±0.84 | 99.04±0.38 | 96.23±0.64 | 92.67±0.65 | 93.23±0.92 |
| | UniTS* | 84.45±0.98 | 83.93±1.21 | 98.27±0.18 | 90.03±0.93 | 84.77±1.14 | 83.41±1.25 |
| | PatchTST* | 88.43±0.44 | 89.87±0.38 | 98.79±0.11 | 94.34±0.40 | 89.73±0.45 | 90.15±0.43 |
| | PatchTST (+PPT)* | **92.33±0.48** | **93.67±0.45** | 99.28±0.10 | 96.83±0.44 | **93.54±0.37** | **93.90±0.56** |
| | PITS* | 87.23±1.04 | 87.18±0.86 | 98.41±0.16 | 91.65±0.77 | 88.09±0.86 | 86.62±0.88 |
| | PITS (+PPT)* | 92.07±0.78 | 92.77±0.63 | **99.28±0.09** | **96.87±0.46** | 92.99±0.70 | 92.83±0.65 |

## 5 EXPERIMENTS

This section evaluates PPT as a pretext task in both self-supervised and supervised learning, demonstrating that enhancing patch order awareness improves model performance. In self-supervised scenarios, we employ linear probing, semi-supervised training, and full fine-tuning (Appendix K). In linear probing, the encoder model $f$ is trained, followed by applying logistic regression to the frozen representation. In semi-supervised training, self-supervised training is performed with model $f$, followed by fine-tuning with a fraction of the dataset and a reduced learning rate (Dong et al., 2024). In the supervised context, we perform an ablation study on the consistency and contrastive learning methods, revealing their individual and combined effects on model performance.

**Implementation of PPT.** We apply PPT to two representative patch-based models: Transformer-based PatchTST (Nie et al., 2022) and linear-based PITS (Lee et al., 2023). We set permutation frequency $\gamma = 1, 2, 3$ for weak shuffle and $\gamma = 10$ or $40$ for strong shuffle. As each dataset differs in length, we adjust the patch size $s$ so that each sequence is divided into 10 to 40 non-overlapping patches. The detailed implementations and configurations are in Appendix D and Appendix F.

**Self-Supervised Baselines.** We make an extensive comparison to leading self-supervised baselines such as TS2Vec (Yue et al., 2022), SimCLR (Tang et al., 2020), TF-C (Zhang et al., 2022), TS-TCC (Eldele et al., 2021), and SimMTM (Dong et al., 2024), as well as patch-based works like TimeMAE (Cheng et al., 2023), TS-GAC (Wang et al., 2023), UniTS (Gao et al., 2024), PatchTST (Nie et al., 2022), and PITS (Lee et al., 2023). For PatchTST, we use the mask-and-

**Table 2: Semi-Supervised Results**. We perform semi-supervised training on the GL task. Models are pre-trained on unlabeled data using self-supervised learning, then fine-tuned on 10% and 1% of labeled data. Best scores are in **red bold**. Full fine-tuning and randomly initialized backbone results are in Appendix K.

| Fraction | Models | Accuracy | F1 score | AUROC | AUPRC | Precision | Recall |
|---|---|---|---|---|---|---|---|
| **10%** | Mixing-up | 92.85±0.69 | 90.44±0.91 | 98.74±0.26 | 94.42±1.14 | 90.59±0.65 | 90.69±0.90 |
| | SimCLR | 84.55±0.78 | 83.60±1.21 | 98.47±0.20 | 91.74±0.83 | 86.78±0.62 | 82.56±1.16 |
| | TS2Vec | 88.12±1.58 | 85.51±1.13 | 96.46±0.78 | 89.37±1.87 | 85.86±0.69 | 85.97±1.71 |
| | TF-C | 83.35±0.48 | 82.73±0.48 | 97.96±0.09 | 87.87±0.36 | 83.95±0.33 | 82.10±0.59 |
| | TS-TCC | **93.69±1.05** | 92.11±0.84 | 99.41±0.19 | 97.36±0.76 | 93.69±0.38 | 91.83±0.83 |
| | TimeMAE | 90.06±2.95 | 91.10±2.54 | 98.77±0.43 | 94.65±2.04 | 91.50±2.46 | 90.83±2.63 |
| | SimMTM | 91.94±0.58 | 91.35±0.53 | 98.95±0.35 | 95.65±0.62 | 91.41±0.44 | 91.40±0.65 |
| | PatchTST | 91.61±0.82 | 92.33±0.89 | 99.35±0.11 | 97.10±0.47 | 92.88±0.80 | 92.47±0.75 |
| | PatchTST (+PPT) | 93.26±1.57 | **93.97±1.40** | **99.50±0.09** | **97.79±0.47** | **94.74±1.23** | **94.27±1.34** |
| | PITS | 85.11±3.78 | 85.67±2.21 | 98.18±0.43 | 89.51±2.63 | 84.61±2.35 | 84.60±2.65 |
| | PITS (+PPT) | 92.47±1.06 | 93.32±0.60 | 99.48±0.12 | 97.28±0.78 | 93.17±0.69 | 93.07±0.46 |
| **1%** | Mixup | 84.82±2.17 | 82.08±2.85 | 97.27±0.53 | 87.48±1.81 | 83.76±2.52 | 81.53±3.34 |
| | SimCLR | 62.61±1.89 | 47.28±4.56 | 90.88±2.03 | 66.05±4.28 | 63.15±9.38 | 51.63±2.92 |
| | TS2Vec | 77.41±1.33 | 75.17±2.85 | 96.17±0.45 | 82.84±1.67 | 79.04±1.10 | 74.64±3.01 |
| | TF-C | 65.34±2.50 | 52.88±4.98 | 91.19±1.59 | 71.15±3.38 | 71.92±4.11 | 52.95±3.65 |
| | TS-TCC | **85.77±1.08** | 83.02±1.16 | 97.82±0.25 | 89.85±1.19 | 86.31±2.00 | 83.04±1.46 |
| | TimeMAE | 76.09±2.01 | 74.63±3.30 | 96.24±0.53 | 80.35±3.35 | 77.58±3.69 | 73.57±3.61 |
| | SimMTM | 78.44±2.20 | 79.48±1.95 | 94.93±0.87 | 82.75±1.34 | 80.66±2.40 | 79.31±1.85 |
| | PatchTST | 80.55±2.29 | 83.26±2.11 | 96.77±1.04 | 86.44±2.83 | 81.50±3.58 | 81.52±3.58 |
| | PatchTST (+PPT) | 84.80±1.68 | **86.92±1.48** | **98.08±0.38** | **90.64±1.90** | **86.88±1.65** | **86.75±1.57** |
| | PITS | 72.41±2.05 | 72.81±4.76 | 95.40±0.60 | 75.92±3.23 | 69.83±3.77 | 70.45±4.71 |
| | PITS (+PPT) | 81.04±1.86 | 83.71±0.95 | 97.68±0.30 | 87.26±1.62 | 81.25±1.45 | 82.05±1.30 |

reconstruct (Mask) pretext task, and for PITS, we use both Mask and complementary contrastive learning (CL) pretext tasks from their original works to differentiate from the use of PPT as pretext task. We use metrics like Accuracy, F1, AUROC, AUPRC, Precision, and Recall, consistent with previous works (Wang et al., 2024b).

**Datasets.** (1) EMOPain (Egede et al., 2020), from the UEA repository (Bagnall et al., 2018), is a pain classification task using a surface electromyographic sensor (sEMG), with 30 channels (2) PTB (Bousseljot et al., 1995) is cardiogram signals (ECG), incorporating 15 channels from 198 users. (3) Gilon (GL) is a large-scale smart insole-based human activity recognition (HAR) task (Kim et al., 2023) with 47,647 instances with 14 channels collected from 72 users. (4) Sleep EEG (Kemp et al., 2000) is a univariate brainwave (EEG) with 371,055 instances. (5) MS-HAR (Morris et al., 2014) has six sensors in an armband with 14,201 instances collected from 93 users. The full details of the datasets are in Appendix E. Here, five random seeds are used to report the results.

## 5.1 SELF-SUPERVISED LINEAR PROBING

**Setup.** In the linear probing setup, PPT is trained self-supervised, optimizing Eq. (3). Strong performance in linear probing indicates a robust learned representation. Following self-supervised training, logistic regression is applied to evaluate the representation (Yue et al., 2022). We used official baseline model implementations, making necessary adjustments (e.g., adapting for multivariate series when only univariate support was available). The hyperparameters for the baselines are in Appendix H.

**Results.** Partial results of three tasks (EMO, PTB, and GL) are in Tab. 1, with full results in Appendix J. Here, we observe that PPT is an effective self-supervised pretext task, providing improved learning representation compared to leading methods. Specifically, PatchTST with PPT is the best-performing model in EMO and GL, while PITS with PPT performs best in PTB. Furthermore, PPT consistently outperforms or matches the original pretext methods in both PatchTST and PITS across all tasks. In particular, PITS with PPT surpasses the original method by five percentage points in AUROC on the PTB task. This performance gain in PTB is supported by a high ACF-COS score of 0.318, indicating that order is essential in the cardiogram data. These results underscore that learning order information provides strong guiding signals, allowing the model to learn interdependencies between patches from both time and channel dimensions, leading to enhanced model performance.

Table 3: **Ablation Study for Supervised Training**. We perform an ablation on the $\mathcal{L}^{CS}$ and $\mathcal{L}^{CT}$ loss components on GL, Sleep EEG, and PTB ECG, using the supervised cross-entropy loss as the baseline for all experiments. The performance metrics are the accuracy on the original un-permuted test set (Original), accuracy on the permuted test set (Permuted), and their Difference (Diff), with higher Original and Diff, and lower Permuted values indicating better performance. A '✓' indicates inclusion, and a '✗' indicates exclusion of the module. The best results are highlighted in **red bold**.

| Dataset Name | | | GL HAR | | | SleepEEG | | | PTB ECG | | |
|---|---|---|---|---|---|---|---|---|---|---|---|
| Models | $\mathcal{L}^{CS}$ | $\mathcal{L}^{CT}$ | Original ↑ | Permuted ↓ | Diff ↑ | Original ↑ | Permuted ↓ | Diff ↑ | Original ↑ | Permuted ↓ | Diff ↑ |
| **PatchTST** (2022) | ✗ | ✗ | $91.6_{\pm3.35}$ | $88.8_{\pm6.01}$ | 2.76 | $61.6_{\pm1.57}$ | $58.5_{\pm1.51}$ | 3.03 | $78.6_{\pm2.16}$ | $76.3_{\pm2.79}$ | 2.28 |
| | ✗ | ✓ | $96.6_{\pm1.00}$ | $89.2_{\pm2.76}$ | 7.33 | $61.8_{\pm1.18}$ | $58.0_{\pm1.03}$ | 3.79 | $78.8_{\pm2.79}$ | $73.7_{\pm1.38}$ | 5.17 |
| | ✓ | ✗ | $97.2_{\pm0.40}$ | $89.0_{\pm4.13}$ | 8.17 | $61.5_{\pm0.61}$ | $\mathbf{57.9_{\pm0.94}}$ | 3.56 | $\mathbf{81.8_{\pm2.48}}$ | $73.5_{\pm2.29}$ | 8.33 |
| | ✓ | ✓ | $\mathbf{97.4_{\pm0.46}}$ | $\mathbf{88.7_{\pm2.59}}$ | **8.65** | $\mathbf{63.5_{\pm0.79}}$ | $58.7_{\pm0.59}$ | **4.69** | $81.4_{\pm2.51}$ | $\mathbf{72.7_{\pm0.91}}$ | **8.71** |
| **PITS** (2023) | ✗ | ✗ | $91.6_{\pm3.32}$ | $85.3_{\pm3.34}$ | 6.30 | $55.4_{\pm1.87}$ | $\mathbf{55.1_{\pm1.85}}$ | 0.32 | $82.0_{\pm6.67}$ | $71.6_{\pm0.66}$ | 10.4 |
| | ✗ | ✓ | $92.8_{\pm4.63}$ | $81.0_{\pm8.30}$ | 11.8 | $56.3_{\pm2.34}$ | $55.6_{\pm2.55}$ | 0.65 | $85.1_{\pm2.98}$ | $68.9_{\pm6.02}$ | 16.2 |
| | ✓ | ✗ | $94.0_{\pm0.68}$ | $87.6_{\pm5.15}$ | 6.40 | $57.4_{\pm1.22}$ | $56.8_{\pm1.10}$ | 0.66 | $84.0_{\pm5.61}$ | $71.3_{\pm0.69}$ | 12.7 |
| | ✓ | ✓ | $\mathbf{96.3_{\pm1.19}}$ | $\mathbf{73.0_{\pm5.29}}$ | **23.3** | $\mathbf{59.3_{\pm0.87}}$ | $57.3_{\pm1.03}$ | **1.95** | $\mathbf{89.5_{\pm1.96}}$ | $\mathbf{65.0_{\pm6.46}}$ | **24.6** |

## 5.2 SEMI-SUPERVISED TRAINING

**Setup.** We evaluated the downstream performance of a self-supervised model through semi-supervised training. This involves pre-training the encoder with unlabeled samples, then adding a fully connected layer, and fine-tuning all parameters using cross-entropy loss (Wang et al., 2024b) with limited labeled training data: 10%, and 1%. We also provide the full fine-tuning results (using 100%) in Appendix K, where we also compare our work with a randomly initialized backbone.

**Results.** Table 2 is the results of semi-supervised training on the GL task. PPT demonstrates superior performance, ranking as the top or second-best model in 10 out of 12 metrics. Given GL's imbalanced nature, metrics beyond accuracy provide more meaningful insights. In both 10% and 1% labeled data scenarios, utilizing PPT as the pretext task leads to substantial performance improvements for PatchTST and PITS compared to their original pretext tasks (mask reconstruction and contrastive learning, respectively). Notably, in the 1% labeled data setup, F1 scores improve by up to 3 percentage points for PatchTST and 11 percentage points for PITS. We hypothesize that PPT pretraining initializes the model in a more order-aware embedding space, enabling it to benefit from self-supervised pretraining even with limited supervised training data.

## 5.3 SUPERVISED TRAINING

**Setup.** We also highlight that PPT can improve supervised training performance as an additional pretext task, with the results in Tab. 3. Here, we selectively add the patch order consistency and contrastive loss terms in Eq. (3) to the cross-entropy loss as the base loss. We evaluated the model using Original, Permuted, and Diff metrics. The Original (↑) metric measures classification accuracy on the standard test set, aiming for higher values. The Permuted (↓) metric assesses accuracy on a previously unseen, strongly permuted ($\gamma = 40$) test set, with lower scores indicating that the model considers order in its decision-making. This method of assessing order awareness through performance degradation has been explored in the time series (Zeng et al., 2023) and video (Yun et al., 2022) domains. Finally, the Diff (↑) represents the difference between the Original and Permuted metric, with more differences signifying more effective learning of order dependencies.

**Results.** The results in Tab. 3 show that incorporating PPT improves model performance (Original) across all six cases, with gains of up to seven percentage points in the PTB task for PITS. Additionally, performance on the Permuted sets decreases in all instances, indicating enhanced patch-order awareness provided by PPT. Notably, the performance drop occurs even when the proposed consistency and contrastive loss terms do not directly penalize logits for permuted patches, unlike methods that optimize to increase uncertainty in out-of-distribution sets (Lee et al., 2017; Yun et al., 2022). The Diff metric also shows increased scores with the addition of each loss, suggesting a synergistic effect when both losses are used. Furthermore, performance drop on the Permuted set is more pronounced in the linear model compared to the Transformer-based model (PatchTST), aligned with findings of existing work (Zeng et al., 2023).

## 5.4 ACF-CoS METRIC FOR TIME SERIES DATASETS

We hypothesize that PPT performs optimally with time series tasks that exhibit ordered information. To test it, we conducted a supervised experiment with PatchTST on tasks from the UEA repository, comparing the performance enhancement of PatchTST+PPT against the baseline of PatchTST that only applies cross-entropy (CE) minimization. As time series models are known to be sensitive to hy-

| UEA Datasets | ACF-CoS ↑ | Acc. Win% (Wins) ↑ | Max CE / Max PPT (Acc) ↑ |
|---|---|---|---|
| *Step Function* (Order ↑) | 0.902 | - | - |
| Cricket | 0.418 | 75.9% (123/162) | 69.4 / **72.7** |
| EigenWorms | 0.289 | 61.1% (99/162) | 47.1 / **54.5** |
| NATOPS | 0.216 | 75.3% (122/162) | 70.0 / **71.7** |
| LargeKitchen. | 0.190 | 54.3% (88/162) | 64.1 / **65.0** |
| GestureMidAirD1 | 0.186 | 48.1% (78/162) | 26.2 / **31.3** |
| GestureMidAirD3 | 0.060 | 37.7% (61/162) | **18.2** / 16.9 |
| *White Noise* (Order ↓) | 0.001 | - | - |

**Table 4:** The ACF-CoS scores for 24 tasks from the UEA Repository: Results for six subsets are shown here. "Acc. Win(%)" is the ratio of the count at which PPT enhances model performance compared to cross-entropy (CE) at various hyperparameter setups. "Max PPT/ Max CE (Acc)." denotes the best accuracy for PPT and CE in all hyperparameters (best in **bold**). Here, an increase in ACF-CoS leads to an improved likelihood of PPT being beneficial. Full results in Appendix L.

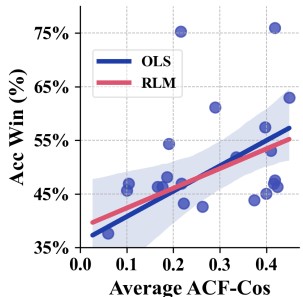

**Figure 5:** The scatter plot for the ACF-CoS score against "Acc. Win". Ordinary Least Square (OLS) and Robust Linear Method (RLM) are fitted to show their correlation.

perparameters (Jin et al., 2024), we systematically conduct experiments with various hyperparameter setups; evaluations are carried out with 162 different hyperparameter configurations, each with three random seeds. To gauge the effectiveness of PPT, we *measure the count* in which PPT **significantly outperforms** CE (cross-entropy), achieving a performance gain exceeding $\geq 1\%$ (expressed as a proportion, not percentage points) of accuracy. The summarized results of 978 experiments (162 configurations × 3 seeds × 2 for both CE and PPT) for each task are in Tab. 4 and Fig. 5, along with the ACF-CoS scores averaged with various patch lengths. The relationship between ACF-CoS scores and performance gains is explored with the Ordinary Least Squares (OLS) (Montgomery et al., 2021) and Robust Linear Model (RLM) using Huber regression (Huber, 1973), resulting in slope coefficients of +0.47 for OLS and +0.37 for RLM, both statistically significant. These findings show a positive correlation between the ACF-CoS score and PPT's capacity to provide additional supervisory signals, which enhances model performance. We discuss our full results in Appendix L.

### 5.5 ADDITIONAL EXPERIMENTS

**Patch Selection Strategy.** We compare various patch permutation strategies under both weak and strong permutations in Appendix C, showing that the random permutation is the most effective. **Hyperparameter Sensitivity Analysis.** We evaluate the performance of PPT with varying patch sizes, $s$, and permutation frequencies, $\gamma$ in both self-supervised and supervised training in Appendix G. **PPT with Time Series Foundation Models.** We applied PPT to GPT4TS (Zhou et al., 2024), a time series foundation model which fine tunes GPT2 for time series (Appendix M). We show that under our patched setup, PPT can enhance performance for time series-based foundation models.

## 6 LIMITATIONS AND DISCUSSIONS

We recognize and discuss several limitations of PPT. First, as detailed in Sec. 5.4, PPT does not enhance performance for tasks where the classification of time series is independent of order information. However, ACF-CoS can help identify such tasks. Second, the credibility of ACF-CoS is influenced by the size of the patch $s$. Although we mitigated it by averaging the ACF-CoS scores in different patch sizes, a more robust metric is needed. Lastly, attempts to apply PPT to time series forecasting tasks tend to yield minimal gains, likely because permuting patches disrupt trend patterns, which are key features in forecasting (Liu et al., 2023; Wu et al., 2021). Designing a universal pretext task for time series is our next research topic.

## 7 CONCLUSION

In this paper, we introduce PPT, a patch order-aware pretext task designed for time series classification models that utilize patch-based representations. We first introduce several strategies to improve patch order awareness in the model by proposing channel-wise patch permutations with varying permutation strength, assisted with consistency and contrastive loss. We show that each loss component plays an important role in enhancing patch order awareness, leading to enhanced model performance. To extend beyond this gain, we develop the novel ACF-CoS metric to identify tasks where PPT can be beneficial, quantifying the importance of order information in a time series. The contribution of this work is a timely response to the growing prevalence of patch-based time series work, addressing the critical challenge of integrating order awareness for improved classification models.

ACKNOWLEDGMENTS

This work was supported by the National Research Foundation of Korea(NRF) grant funded by the Korea government(MSIT)(RS-2023-00277383) and Institute of Information & communications Technology Planning & Evaluation(IITP) grant funded by the Korea government(MSIT) (No.RS2020-II201336, Artificial Intelligence Graduate School Program(UNIST)). The authors are grateful to Seok-Ju Hahn and Yoontae Hwang for their helpful discussions during the initial ideation phase. Special thanks go to Hyunwoo Seo, Yeonjoo Kim, Gyeongho Kim, Isu Jeong, Kyu Hwan Lee, Ahin Lee, and the EAI Lab members for their feedback on the manuscript. The authors thank the anonymous reviewers for their constructive feedback during the discussion phase.

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

# A  FEATURE ABLATION

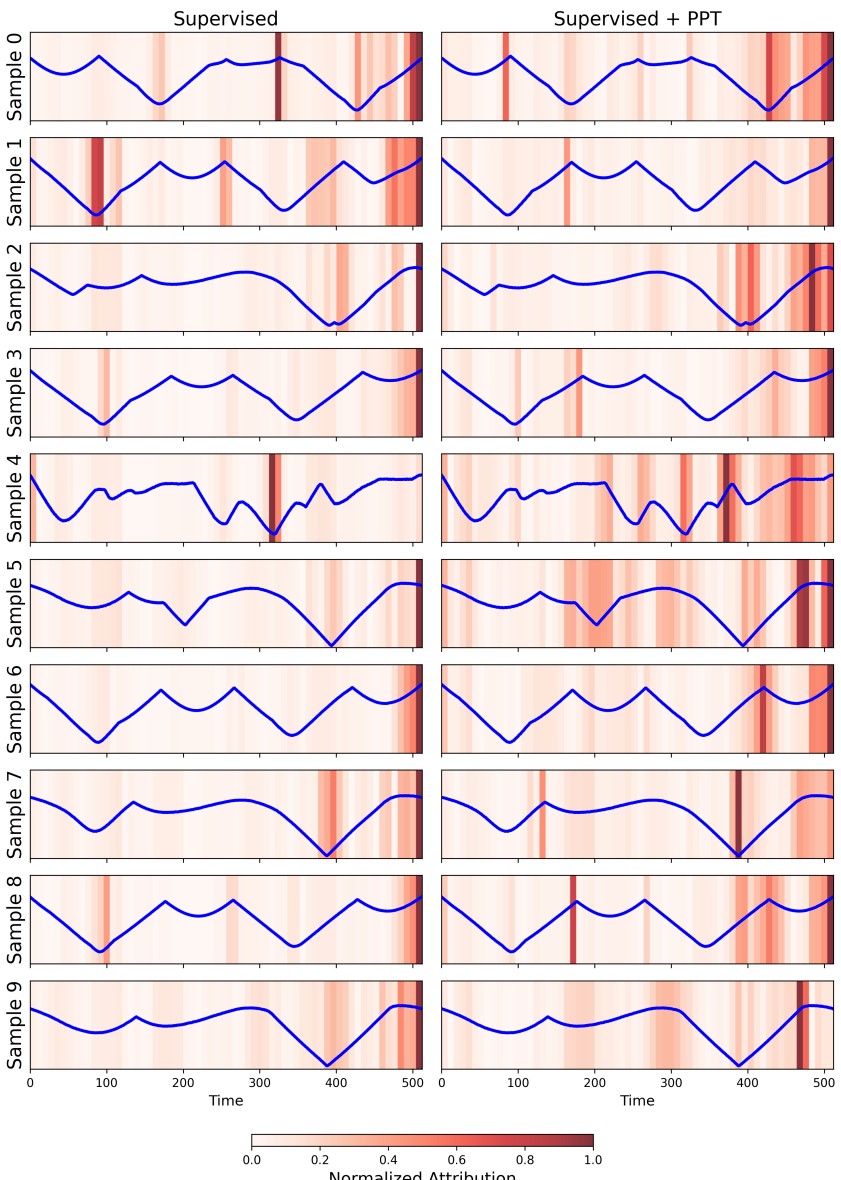

**Figure 6:** The figure presents ten randomly selected attribution results from the ShapesAll task under supervised training conditions. We compare two approaches: (1) the baseline method that uses only cross-entropy (CE) loss, and (2) our proposed method that combined PPT with cross-entropy loss. We applied the FeatureAblation method to assess the importance of patches in model performance. The ShapesAll task was chosen for visualization due to its high ACF-CoS score, indicating that temporal order is important in this dataset. Higher ACF-CoS scores suggest that order information is crucial for the task. We observe that incorporating PPT leads to better attribution scores in meaningful time segments of the data compared to using CE alone. For instance, in Sample 4, we observe that the attributions are evenly distributed and the scores are high at the inflection points for PPT, while the baseline only focuses on a single time segment.

---

[1]We used the Captum (https://captum.ai/) library for FeatureAblation.

## B    EXTENDED RELATED WORKS

**Order-Aware Self-Supervised Learning.** Taking advantage of inherent order information within various data modalities has proven to be an effective pretext task, enhancing model performance across domains. In the vision domain, structured data like pixels or patches has led to the development of order-aware tasks, such as relative position prediction Doersch et al. (2015), jigsaw puzzle Noroozi & Favaro (2016), and rotation Gidaris et al. (2018) for convolutional networks. More recently, Vision Transformer (ViT) models Dosovitskiy et al. (2020), which segment images into a grid of patches, have shown to be effective in modeling long-range dependencies but fail to incorporate order dependency between patches Naseer et al. (2021). As such, for ViTs, predicting the absolute positions of image patches by masking Zhai et al. (2022) and reconstructing positions of patches Wang et al. (2024a) have been proposed. Similarly, in video analysis, leveraging the temporal sequence of frames is crucial, as it encourages models to grasp frame dynamics over static features Yun et al. (2022). Similarly, in natural language processing, the core assumption behind masked language modeling is that the sequential order of words in a sentence contains critical cues for language understanding Devlin et al. (2018); Mikolov et al. (2013). Thus, utilizing structural order is key to designing meaningful pretext tasks across various data modalities.

**Structure-Aware Learning in Time Series.** In time series analysis, several methods have their design principle based on the structural relationship within time sequences. SASA Cai et al. (2021), in time series domain adaptation, focuses on extracting domain invariant representations by analyzing the causal structure among time and channel dimensions. In self-supervised learning for time series, TNC Tonekaboni et al. (2021) addresses the temporal consistency of clinical time series by defining a neighborhood function to extract positive samples for contrastive learning. Similarly, CLOCS Kiyasseh et al. (2021) deals with the temporal and spatial (channel) invariance of ECG applications based on the assumption that the information from the same temporal events is conserved in different time sequences and channels. TS2Vec Yue et al. (2022) is an universal time series representation method emphasizing contextual consistency. TF-C Zhang et al. (2022) utilizes both the time and frequency domain and proposes time-frequency consistency architectures. SimMTM Dong et al. (2024) constructs multiple masked series, where they consider these series as close neighbors of the original time series. TS-TCC Eldele et al. (2021) focuses on transformation consistency, generating two views of the time series data using weak (jittering and scaling) and strong (permuting and jittering) augmentations. Then, an autoregressive model is used to construct a context vector for temporal contrasting between the two samples. This concept was expanded into semi-supervised learning (*i.e.* CA-TCC Eldele et al. (2023)) and graph-structured learning (*i.e.* TS-GAC Wang et al. (2023)). Inspired by TS-TCC, PPT introduces a channel-wise patch permutation approach with varying strengths determined by the permutation frequency, termed 'weak' and 'strong' permutation. However, while the main objective of TS-TCC is to bring the weak and strong representations closer, our objective is the opposite, which is to widen the distance between weak and strong, while placing the weak closer to the non-permuted 'original' set (Sec. 3.2). Moreover, unlike TS-TCC's random segment permutations Um et al. (2017) made on the fly, PPT gives more control over the strength of permutation, which enables a time and feature-wise consistency and contrasting mechanism (Sec. 3.3). In addition, PPT is efficient due to its use of a permutation bank (*i.e.* only the permuted indices are stored) and effective because it continuously exposes the model to a diverse array of permuted samples. As such, while PPT utilizes a similar augmentation strategy from TS-TCC, it significantly diverges in its focus for contrasting, making it a novel and effective patch-based pretext task for time series.

## C   COMPARISON BETWEEN PATCH SELECTION STRATEGY

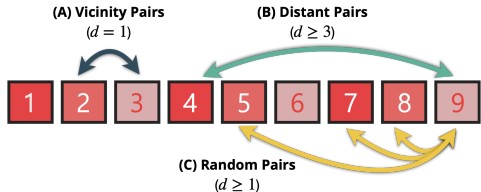

**Figure 7:** The figure illustrates the patch selection strategy utilized in the channel-wise patch permutation strategy, where two patches are selected from the same channel, and their positions are swapped. The selection method varies according to the strategy adopted, which is quantified by the distance $d$ between the patch pairs. **(A) Vicinity Pairs** ($d = 1$): The adjacent patches with distance $d = 1$ are swapped. **(B) Distant Pairs** ($d \geq 3$): Patches separated by at least three patches are swapped. **(C) Random Pairs** ($d \geq 1$): This strategy includes a combination of both vicinity and distant pairs.

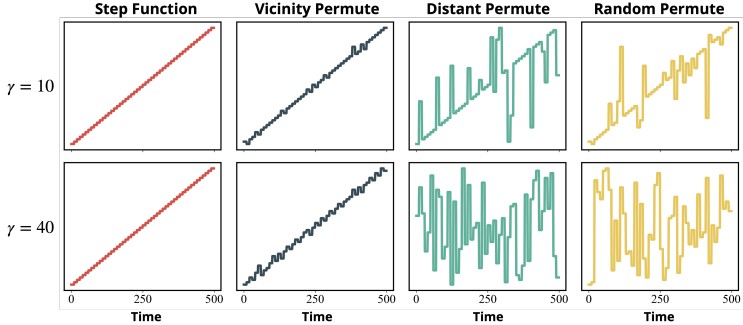

**Figure 8:** We present visualizations of the results obtained from permuting a step function using various permutation strategies: Vicinity, Farthest, and Random. The permutation frequency hyperparameter, $\gamma$, was set to $\gamma = 10$ for the first row and $\gamma = 40$ for the second row.

| | | Gilon HAR | | PTB | | |
| --- | --- | --- | --- | --- | --- | --- |
| **Strong** | **Weak** | **PatchTST ↑** | **PITS ↑** | **PatchTST ↑** | **PITS ↑** | **Average Rank ↓** |
| Vicinity | Vicinity | 97.2±0.55 | 96.0±0.76 | 80.2±2.89 | 86.2±2.19 | 5.25 |
| Vicinity | Distant | **97.5±0.51** | 96.1±1.16 | 79.0±3.29 | 85.8±1.40 | 5.25 |
| Vicinity | Random | 97.3±0.57 | 96.4±0.82 | 78.4±1.86 | 86.9±1.57 | **4.75** |
| Distant | Vicinity | 96.7±0.84 | 95.7±1.01 | **83.1±4.79** | 86.1±2.00 | 6.00 |
| Distant | Distant | 96.7±0.69 | 96.6±0.52 | 78.4±3.45 | **88.6±0.93** | 5.00 |
| Distant | Random | 96.5±1.72 | **96.7±0.41** | 80.6±3.27 | 84.7±4.22 | 5.75 |
| Random | Vicinity | 96.9±1.07 | 96.0±0.71 | 79.6±3.83 | 87.8±1.64 | 5.50 |
| Random | Distant | 97.0±0.90 | **96.8±0.61** | 78.2±3.95 | 86.3±2.52 | 5.00 |
| Random | Random | **97.4±0.46** | 96.3±1.19 | **81.4±2.51** | **89.5±1.96** | **2.50** |

**Table 5:** The classification accuracy from a supervised experiment with varying permutation strategies are given. Here, the strong permutations were implemented using $\gamma = 40$, while weak permutations used $\gamma = 1$. The highest performing value is in **red bold**, and the second highest in **black bold**. We calculated the rank for each dataset and model and averaged those ranks to obtain **Average Rank**, where a lower score indicates better performance. Overall, the Random-Random combination yielded the best results.

# D    CHANNEL-WISE PATCH PERMUTATION

---

**Algorithm 1** Pseudocode of Channel-Independent Patch Permutation in PyTorch style.

---

```
1   # B: batch size; C: channel number; T: Original Time Length
2   # PD: Patch Dimension; PN: Patch Numbers (Number of Patches);
3
4   # Print original shape of X
5   print(X.shape) # -> (B, C, T)
6
7   # Reorder X into non-overlapping patches.
8   X = rearrange(X, "B C (PD PN) -> B PD C PN")
9
10  # Sample a random value (key)
11  key = torch.randint(0, 1000, (1,))
12
13  # Permute Bank is a dictionary containing permuted indexes.
14  shuffled_idx = permute_bank[key]
15
16  # Print shuffled_idx.
17  print(shuffled_idx) # -> (C, PN)
18
19  # Expand the shape of idx to match the shape of X
20  expanded_shuffled_idx = shuffled_idx.expand(B PD C PN)
21
22  # Rearrange original patch based on expanded_shuffled_idx
23  X_shuffled = torch.gather(X, 3 expanded_shuffled_idx)
24
25  # Print X_shuffled shape
26  print(X_shuffled.shape) # (B PD C PN)
27
28  # Rearrange shape back to its original
29  X_shuffled = rearrange(X, "B PD C PN -> B C PD PN")
30
31  return X_shuffled
```

---

rearrange: rearrange elements of a tensor according to a pattern; `torch.gather`: gather values along an axis specified by index; `expand`: repeat the tensor along specified dimensions to match the given shape.

---

**Implementation.** The channel-wise patch permutation strategy is efficiently implemented, as outlined in the PyTorch-style pseudocode in Algorithm 1. Prior to model training, a permutation bank is constructed, which is a dictionary storing permuted patch indices. For example, in the case of a 32-patch permutation, instead of retaining the actual time series data for the patches, only the indices of these patches are stored. 1000 different variations of these permuted indices are stored to provide a wide range of permutations during model training. During model runtime, a random key is generated, and the matching permuted indices is retrieved and the patches are repositioned accordingly.

**Time.** Building such a dictionary for a task with 14 channels, 32 patches, permutation frequency $\gamma = 40$, with 1000 possible variations takes only 2.6 seconds on an Intel(R) Xeon(R) Gold 6226R CPU @ 2.90GHz. Permuting a batch of 512 instances from this task takes merely 0.018 seconds on modern GPU hardware like the NVIDIA RTX A6000. This approach ensures that during training, the model is exposed to a broad spectrum of permuted orders of the dataset, while being fast, light-weight, and efficient.

**Weak and Strong Permute Strategy.** For our implementation of PPT, we built multiple permutation dictionaries with varying permutation frequency $\gamma$ to support both weak and strong permutation strategies. For weak permutations, we built three dictionaries for $\gamma = 1, 2, 3$. For strong permutations, we select a lower bound $\gamma = l$ and construct dictionaries for $\gamma = l, l+1, l+2$. For each training batch, one of these dictionaries from both weak and strong is randomly selected for use. This method further enhances the variety of permutations the model encounters during training.

# E  DATASETS

| Dataset | Task Type | #Samples | Time | #Users | # Channels | # Class |
|---|---|---|---|---|---|---|
| MS | HAR | 14,201 | 200 | 93 | 6 | 10 |
| Gilon | HAR | 47,647 | 160 | 72 | 14 | 7 |
| EMOPain | sEMG | 1,143 | 200 | 30 | 30 | 3 |
| PTB | ECG | 62,370 | 300 | 198 | 15 | 2 |
| Sleep | EEG | 371,055 | 178 | 82 | 1 | 5 |

**Table 6:** Details of the datasets used in our main experiments

## E.1  RATIONALE FOR DATASET SELECTION

The selection of the five datasets used in our main experiment was based on three major reasons: (1) public accessibility, (2) a sufficient number of samples for generalization, and (3) a diverse representation of time series tasks. All datasets utilized in our experiments are freely accessible, except for the Gilon task, which is available upon request and only for research purposes. Moreover, each time series task in our study includes more than 1,000 samples, with four datasets containing more than 10,000 instances each, supporting effective generalization. In addition, the chosen tasks span a range of applications in time series analysis, including Human Activity Recognition (HAR), Electroencephalography (EEG) for brain wave analysis, electrocardiography (ECG) for cardiogram monitoring, and Surface Electromyography (sEMG) for muscle activity. This diversity demonstrates the applicability of PPT to a wide range of tasks. We believe that our work would be particularly beneficial in fields where sequence order is crucial, such as in biosignal analysis.

## E.2  DATASET DESCRIPTION

**MS HAR (Morris et al., 2014).** The MS HAR task is a human activity recognition (HAR) dataset using data collected from 114 users who performed 72 different gym activities, measured with a smart armband. The armband incorporates six sensor channels: three accelerometers (x, y, z axes) and three gyroscopes (x, y, z axes). To address data imbalance among the classes, we selected 10 activities from the original 72 classes. These include Bicep Curl (0), Biceps Curl with Band (1), Jump Rope (2), Plank (3), Pushups (4), Squat (5), Squat with Hands Behind Head (6), Squat Jump (7), Walk (8), and Walking Lunge (9). The raw dataset is available for access at (https://github.com/microsoft/Exercise-Recognition-from-Wearable-Sensors). We will provide the preprocessed dataset through GitHub.

**Gilon HAR (Kim et al., 2023).** The Gilon dataset is developed for HAR and includes data collected from 72 participants performing seven different gym activities. The data was gathered using a smart insole that features 14 sensor channels, including accelerometers (x, y, z axes) and four force-sensitive resistors (FSRs) for each foot (right and left). We used the original five-fold split given in the original paper. The dataset can be accessed with request (https://github.com/Gilon-Inc/GILON-MULTITASK-DATASET) and is publicly available for research purposes.

**EMOPain (EMG) (Egede et al., 2020).** EMOPain is a pain classification task using surface electromyographic (sEMG) sensor to classify different pain levels (0: Healthy, 1: Low-level pain, 2: High-level pain). There are total of 30 sensor channels positioned at strategic anatomical positions. The dataset can be accessed through the UEA time series classification repository (Bagnall et al., 2018).

**PTB (ECG) (Bousseljot et al., 1995).** The PTB ECG dataset comprises cardiogram signals from 294 patients, including healthy subjects. We utilized a preprocessed subset of this dataset from the study by (Wang et al., 2024b), which includes data from 198 patients. This subset redefines the original task into a binary classification problem, distinguishing between Myocardial Infarction and healthy controls. We adopted the same data splits as specified in their study. The preprocessed dataset can be accessed from (https://github.com/DL4mHealth/COMET).

**Sleep (EEG) (Kemp et al., 2000).** The Sleep EEG is a univariate brainwave signal from Electroencephalography (EEG) recordings, where the task is to classify one out of five sleeping patterns: Wake, Non-rapid eye movement (N1, N2, N3) and rapid eye movement (REM). We utilized the preprocessed

dataset from (Zhang et al., 2022) and the dataset can be accessed from (https://github.com/mims-harvard/TFC-pretraining).

# F    IMPLEMENTATION OF PPT

## F.1    HARDWARE AND SOFTWARE

All experiments were conducted on two types of GPU: NVIDIA RTX 3090-24GB and NVIDIA RTX A6000-48GB. We used the PyTorch 2.0.1 and PyTorch Lightning 2.1.2 frameworks. All experiments can be completed on a single GPU mentioned above.

## F.2    TIME AND MEMORY COST OF PPT

We conducted additional experiments to measure the time and memory complexity of PPT in model training. Specifically, we conducted experiments on the PTB task for self-supervised linear probing and have compared our method against mask-based and contrastive-based approaches using the same backbone model (*e.g.* PatchTST).

In summary, PPT employs additional auto-regressive models for constructing context vectors in the training process, which is the main overhead in speed and memory consumption compared to previous pretext task methods. However, in the inference stage, these auto-regressive models can be discarded. In terms of time complexity, mask-based approaches took 70.3 seconds for the whole training, contrastive-based approaches took 92.9 seconds, while PPT took 139.5 seconds. This increased training time is due to the additional computations required for auto-regressive models constructing context vectors. Regarding memory complexity, the mask-based approach had a total of 79.7K parameters, the contrastive-based approach had 79.3K parameters, and PPT had 212K parameters. The higher number of parameters in PPT is primarily attributed to the use of auto-regressive models.

Despite the increased computational overhead, we believe that the improved performance achieved by PPT, as demonstrated in our experimental results, justifies the use of this technique. We acknowledge that the current implementation of PPT has room for optimization, and we propose that exploring more efficient model designs for constructing context vectors (e.g., reducing representation dimensions, and re-using the auto-regressive models) would mitigate the computation burden.

## F.3    AUTO-REGRESSIVE MODELS

To generate the context vectors in consistency and contrastive learning, we used a single-layer unidirectional LSTM that takes the patch sequences as input. The final cell state of the LSTM serves as the context vector, effectively summarizing the sequence of patches.

For consistency learning, we assigned pseudo labels: 1 for sequences of patches from the original sequence and 0 for sequences of patches from a strongly shuffled sequence. Initially, the LSTM is unable to recognize the correct ordered sequence. However, through end-to-end training alongside the backbone patch model, the model learns to discriminate between the original and strongly shuffled patch sequences. Consistency learning is applied along both the time and channel dimensions. For the time dimension, the module learns to recognize the correct temporal order of patches. For the channel dimension, the module learns the cross-channel relationships of patches. As shown in Fig. 1, distinct patterns co-occur along the channel dimension, and learning this cross-channel relationship has been shown to enhance model performance.

For contrastive learning, we collected the context vectors from the original, weakly permuted, and strongly permuted patch sequences. Intuitively, the weakly permuted sequence should have high similarity with the original sequence, while the strongly permuted sequence should be far from both the original and weakly permuted sequences in the representation space. To achieve this, we employed the InfoNCE loss for contrastive training.

The combination of consistency and contrastive learning allows the model to capture both the temporal and cross-channel relationships of patches, leading to improved performance in time series classification tasks. By learning to recognize the correct order and relationships between patches, the model can better understand the underlying patterns and structure of the time series data.

## F.4 SEQUENTIAL REPRESENTATION OF PATCHES AND HEAD TYPE

The latent data representation for patch-based models consists of encoded patch representations, differing from models that provide instance-wise representations. Previous studies have processed these representations by flattening (Nie et al., 2022) or averaging Lee et al. (2023) them to form an instance-wise format. However, these methods may not adequately capture the sequential dependencies between patches. To better capture these dependencies, we utilized a single layer LSTM model (Hochreiter & Schmidhuber, 1997) to generate instance-wise or sequential representations from patches. The patch representation is specified as $\mathbb{R}^{C \times PD \times PN}$, where $C$ is the number of channels, $PD(d_h)$ is the dimension size of the patch, and $PN$ is the number of patches per channel. Initially, we averaged across the channel dimension to produce a patch representation of shape $\mathbb{R}^{PD \times PN}$. This averaged representation serves as the input to the LSTM model, treating $PN$ as the sequence length. In all experiments, we employ the context vector from the LSTM as the instance-wise representation.

## F.5 HYPERPARAMETERS FOR SUPERVISED LEARNING

The hyperparameter configurations for supervised training are in Tab. 7. We utilized the official implementations of both PatchTST and PITS, adopting the channel-independent configuration as outlined in its respective studies. For PatchTST, three layers of transformer encoders were used. A slightly high learning rate was found to improve the performance in PITS. Patch sizes were determined task specifically, with the aim of 10 to 40 patches per channel. No learning rate scheduling or weight averaging techniques were used for PPT. The training was terminated when the validation loss did not decrease for more than two epochs. The temperature $\tau$ for InfoNCE was set to 0.1.

| Model | Task | Batch Size | Permutation $\gamma$ | Learning Rate | Patch Size | $d_h$ | $n_{heads}$ |
|---|---|---|---|---|---|---|---|
| PatchTST | Gilon | 128 | 10 | 0.002 | 5 | 128 | 4 |
| PatchTST | SleepEEG | 256 | 10 | 0.002 | 5 | 128 | 8 |
| PatchTST | PTB | 128 | 10 | 0.002 | 10 | 128 | 8 |
| PITS | Gilon | 128 | 10 | 0.02 | 10 | 128 | - |
| PITS | SleepEEG | 256 | 10 | 0.02 | 5 | 256 | - |
| PITS | PTB | 256 | 10 | 0.02 | 10 | 256 | - |

**Table 7:** Supervised Training Hyperparameter Configurations.

## F.6 HYPERPARAMETERS FOR SELF-SUPERVISED LEARNING

The hyperparameter configuration for self-supervised linear probing is in Tab. 8. For full fine-tuning, the hyperparameters are identical except for the learning rate, which is reduced to one-tenth of that used in linear probing. In self-supervised training, we do not use a separate validation set; instead, the model is trained for a predetermined number of steps. Subsequently, we fitted a logistic regression model using the methodology from TS2Vec (Yue et al., 2022).

| Model | Task | Batch Size | $\gamma$ | Learning Rate | Patch Size | $d_h$ | Train Steps |
|---|---|---|---|---|---|---|---|
| PatchTST | Gilon | 32 | 40 | 0.002 | 5 | 128 | 1000 |
| PatchTST | EMO | 32 | 40 | 0.002 | 10 | 256 | 1000 |
| PatchTST | PTB | 32 | 40 | 0.001 | 6 | 64 | 4000 |
| PatchTST | MS | 32 | 40 | 0.001 | 4 | 256 | 1000 |
| PITS | Gilon | 32 | 40 | 0.002 | 10 | 128 | 4000 |
| PITS | EMO | 32 | 40 | 0.01 | 10 | 128 | 1000 |
| PITS | PTB | 32 | 40 | 0.01 | 6 | 512 | 4000 |
| PITS | MS | 32 | 40 | 0.002 | 4 | 512 | 4000 |

**Table 8:** Self-Supervised Learning Hyperparameter Configurations. We fixed the temperature $\tau$ to 0.1

## G    HYPERPARAMETER SENSITIVITY

### G.1    SUPERVISED LEARNING SENSITIVITY ANALYSIS

We performed a hyperparameter sensitivity analysis in both supervised and self-supervised learning. Specifically, we are interested in how the size of patches and the frequency of permutations affect the performance of the model in supervised learning. We report the results for the Gilon task in Fig. 9. Here, Gilon task has a time length of 160, as such we used patch lengths of 5, 10, 16, and 20. We set the strong permutation frequency between 5, 10, 20, and 40. We observe that the patch size is a key hyperparameter in obtaining optimal performance for PatchTST in the supervised setup, where as PITS was robust across various patch sizes. In general, increasing the permutation frequency tends to enhance model performance. However, a high permutation frequency does not guarantee improved performance in all cases. We hypothesize that, beyond a certain threshold, further permutations do not significantly alter the outcome.

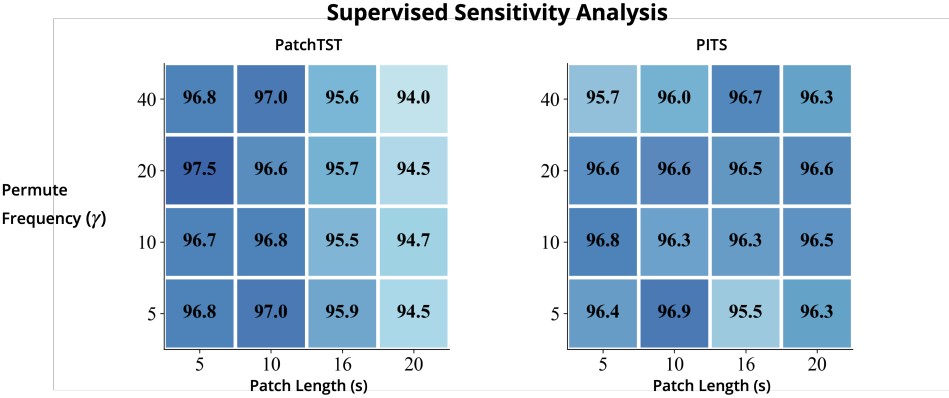

**Figure 9:** A sensitivity analysis of two hyperparameters on supervised Gilon task were conducted: patch length and permute frequency. The x-axis indicates the patch length and the y-axis represents the intensity of the permutation frequency. For PatchTST, we observe an increase in performance with a smaller patch size. In contrast, PITS displays mixed results, maintaining consistent performance across different hyperparameters.

### G.2    SELF-SUPERVISED LEARNING HYPERPARAMETER SENSITIVITY ANALYSIS

We conducted a hyperparameter sensitivity analysis for the PTB (ECG) task within a self-supervised learning setup, with the results in Fig. 10. Unlike in supervised learning, PITS demonstrated a greater sensitivity to patch size than PatchTST. Based on these observations, we conjecture that the sensitivity to patch size and permutation frequency depend on the specific learning setup and the type of task to which it is applied.

### G.3    TEST ON PERMUTED DATASET

We evaluated the performance degradation of the supervised PITS model on a permuted test set (PTB task), detailed in Fig. 11. This figure expands upon Tab. 3, which presented results for a test set permuted at a frequency of $\gamma = 40$. Here, we provide comprehensive results for test sets permuted at frequencies $\gamma = 5, 10, 20, 40$. Additionally, $\gamma = 0$ represents the unaltered original test set. The figure shows results for the three distinct permutation strategies—farthest, random, and vicinity—with each strategy described in Appendix C. For example, the farthest permuted set is the test set that has been permuted with the "farthest" strategy.

In general, we observe that incorporating PPT enhances model performance on the original unaltered test set ($\gamma = 0$), while having a degraded performance on the permuted test sets ($\gamma = 5, 10, 20, 40$). We also observe that the extent of degradation varies with the permutation strategy, being more pronounced under the farthest and random strategies. This variation is intuitive, as the vicinity permutation strategy maintains more of the original sequential information, as illustrated in Fig. 8.

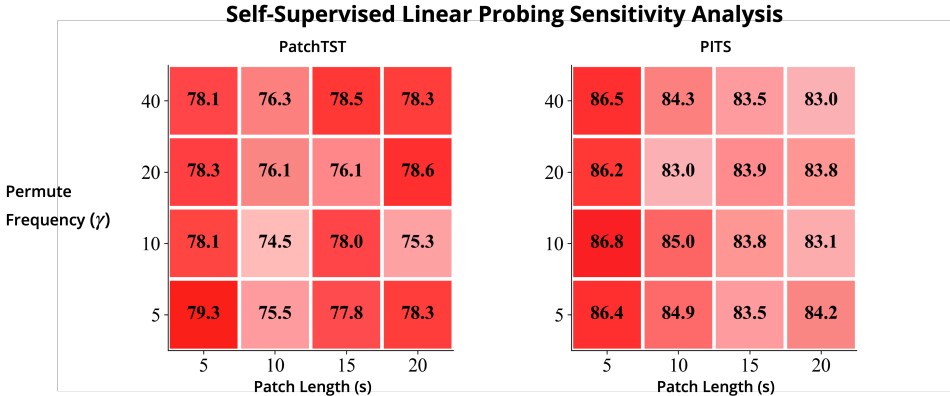

**Figure 10:** A sensitivity analysis on self-supervised PTB (ECG) task were conducted. Unlike the supervised setup, we now observe that PITS is more sensitive to patch size.

**Figure 11:** We trained PPT with Random-Random permutation strategy with the strong permutation frequency set to $\gamma = 40$. The model trained with PPT and without PPT was evaluated on a permuted test set, where the permutation of the test set was set to $\gamma = 5, 10, 20, 40$.

**A note on the drop in performance for permuted test sets.** For first viewers, it is possible that the drop in model performance on the permuted test set seems trivial or insignificant. However, we emphasize that such a drop occurred even though there was no direct penalization of the logits from the permuted patches, unlike related works that directly penalize logits for uncertain samples (Lee et al., 2017) in out of distribution detection problems. During the training of PPT, the permuted sets were used to calculate the consistency and contrastive losses, but no logits from the patch sequences were directly penalized. As such, the degradation in model performance is solely due to the learned representation, which can be considered significant.

## G.4 LEARNABLE AND FIXED LAMBDA

| | PatchTST | | | PITS | | |
|---|---|---|---|---|---|---|
| **Rank** | **F1** | $\mathcal{L}^{\mathbf{CS}}$ | $\mathcal{L}^{\mathbf{CT}}$ | **F1** | $\mathcal{L}^{\mathbf{CS}}$ | $\mathcal{L}^{\mathbf{CT}}$ |
| - | **88.67**±0.95 | learned | learned | **91.24**±0.72 | learned | learned |
| 0 | 88.57±0.78 | 1 | 0.1 | 91.20±0.66 | 0.1 | 1 |
| 1 | 88.39±0.81 | 0.5 | 0.1 | 91.00±0.79 | 0.1 | 0.1 |
| 2 | 87.63±2.30 | 0.5 | 1 | 90.99±0.46 | 2 | 0.1 |
| 3 | 87.50±1.02 | 1 | 0.2 | 90.99±1.03 | 0.5 | 0.2 |
| 4 | 87.46±1.58 | 0.2 | 0.2 | 90.98±0.54 | 0.1 | 2 |
| 5 | 87.32±0.62 | 2 | 0.5 | 90.89±0.86 | 1 | 1 |
| 6 | 87.19±1.63 | 0.1 | 0.2 | 90.84±1.21 | 1 | 0.2 |
| 7 | 87.16±1.23 | 0.5 | 2 | 90.82±0.48 | 0.2 | 0.1 |
| 8 | 87.08±1.93 | 0.1 | 1 | 90.80±1.04 | 0.5 | 2 |
| 9 | 87.04±0.70 | 0.2 | 0.1 | 90.78±0.36 | 2 | 0.5 |
| 10 | 86.96±1.71 | 0.2 | 1 | 90.76±1.18 | 1 | 2 |
| 11 | 86.91±0.66 | 2 | 0.1 | 90.68±0.66 | 2 | 2 |
| 12 | 86.88±1.94 | 0.2 | 2 | 90.66±0.27 | 0.2 | 0.5 |
| 13 | 86.83±0.81 | 0.5 | 0.5 | 90.63±0.42 | 0.1 | 0.2 |
| 14 | 86.76±1.28 | 1 | 2 | 90.54±0.75 | 0.2 | 1 |
| 15 | 86.68±1.84 | 0.1 | 2 | 90.51±0.23 | 0.5 | 0.5 |
| 16 | 86.55±0.97 | 2 | 2 | 90.48±0.71 | 0.5 | 1 |
| 17 | 86.54±1.68 | 1 | 1 | 90.45±0.54 | 0.5 | 0.1 |
| 18 | 86.53±0.72 | 1 | 0.5 | 90.43±0.96 | 1 | 0.5 |
| 19 | 86.27±1.22 | 0.2 | 0.5 | 90.28±0.82 | 1 | 0.1 |
| 20 | 86.16±0.98 | 2 | 0.2 | 90.04±0.46 | 0.2 | 2 |
| 21 | 86.10±0.53 | 0.5 | 0.2 | 90.02±1.05 | 2 | 1 |
| 22 | 86.07±0.33 | 2 | 1 | 90.02±1.14 | 2 | 0.2 |
| 23 | 86.07±1.72 | 0.1 | 0.5 | 89.95±1.01 | 0.2 | 0.2 |
| 24 | 85.61±1.22 | 0.1 | 0.1 | 89.23±1.38 | 0.1 | 0.5 |

**Table 9:** Performance comparison between PITS and PatchTST based on fixed loss coefficient

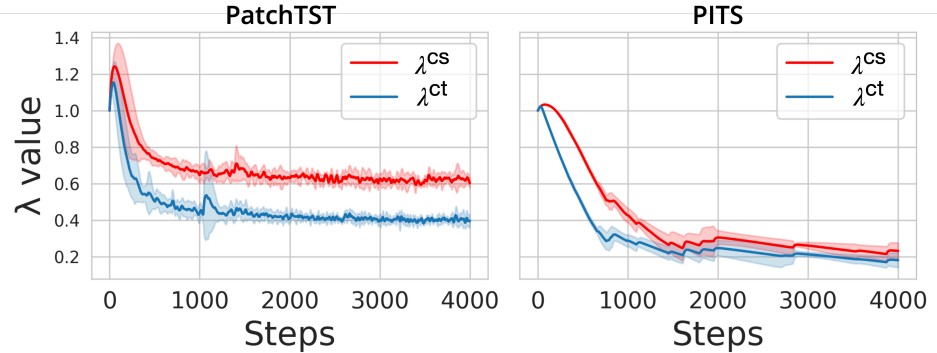

**Figure 12:** We visualized how the learnable parameters for $\mathcal{L}^{\mathbf{CS}}$ and $\mathcal{L}^{\mathbf{CT}}$ changes over the course of training.

To demonstrate the effectiveness of using a learnable $\lambda$ instead of a fixed $\lambda$, we conducted multiple fixed-$\lambda$ experiments and compared them with the results of the learnable $\lambda$. Briefly, we show that the learned $\lambda$s are the best performing compared to the fixed hyperparameter setup.

We conducted a grid search between $\mathcal{L}^{\mathbf{CS}} = 0.1, 0.2, 0.5, 1.0, 2.0$ and $\mathcal{L}^{\mathbf{CT}} = 0.1, 0.2, 0.5, 1.0, 2.0$, resulting in 25 different combinations. We performed our experiments on the PatchTST (Transformer) and PITS (Linear) model with the PTB dataset, and reported the results in Tab. 9.

We show that the learnable hyperparameters are comparable to the best performing fixed-hyperparameters that are manually searched, showcasing that the learnable hyperparameter setup can significantly reduce hyperparameter tuning while leading to optimal performance.

## H  BASELINE IMPLEMENTATIONS FOR SELF-SUPERVISED LEARNING

For all baseline experiments in self-supervised linear probing, we standardized the batch size to 32. We conducted a grid search to determine the optimal learning rate, exploring values between $[0.001, 0.002]$, and for the maximum number of steps, ranging between $[1000, 4000]$, and reported the best outcomes. During full fine-tuning, we lowered the learning rate to the one-tenth of the learning rate used in linear probing. Here, we did not implement learning rate scheduling in any part of our study. For patch-based methodologies (PatchTST, PITS, TimeMAE, TS-GAC), we aimed to set a patch size that ensured the number of patches per sequence fall in between 10 to 40.

**PatchTST** (Nie et al., 2022) is an early work utilizing patch representations in time series forecasting, employing a basic transformer architecture. We selected PatchTST as our main encoder backbone, expecting that PPT could be generalized to other transformer variants. Specifically, the work proposes to use a channel-independent encoder transformer network. We used channel-independent encoder architecture with sine-cosine positional encoding from the original implementation. Across the study, we employed three transformer layers. The RevIN normalization (Kim et al., 2021) used in the original paper was not used as our focus was on time series classification. For supervised training, we utilized the representations from the last transformer layer, inputting them into an LSTM-based classifier head to generate instance-wise representations. In the self-supervised training, we averaged the representations from all transformer layers and used a single-layered, non-bidirectional LSTM to process the patch representations. Patches were averaged along the channel dimension, where the patch indexes were considered as sequences. The final context vector was then employed in our analyses. In self-supervised training, we used patch size of 4 for MS HAR, 5 for Gilon, 6 for PTB, and 10 for EMOPain. We used a batch size of 32 and learning rate of 0.002 for self-supervised linear probing and 0.0002 for full fine-tuning. For the mask and reconstruction pretext task, we employed the original implementation from (https://github.com/yuqinie98/PatchTST).

**PITS** (Lee et al., 2023). A major discussion in the time series domain is the use of MLP based models instead of the transformer architectures. Surprisingly, simple MLP based models tend to outperform transformer architectures, which has been actively discussed in (Zeng et al., 2023). We employed the MLP-based PITS architecture, following the official implementation (https://github.com/seunghan96/pits), to demonstrate that PPT is also effective with MLP models. For self-supervised training, we used the following patch sizes: 4 for MS HAR, 10 for Gilon, 6 for PTB, and 10 for EMOPain. Similarly to PatchTST, we implemented an LSTM-based classifier head to obtain instance-wise representations.

**Mixing-up** (Wickstrøm et al., 2022) is a contrastive learning-based method for time series inspired by the mixup (Zhang et al., 2017) data augmentation strategy. This approach blends two different mini-batch samples to create an augmented sample. A contrastive loss is applied to the augmented sample, with each mini-batch treated as a positive sample weighted by the mixing coefficient. We used the official implementation from (https://github.com/Wickstrom/MixupContrastiveLearning) and set the hidden dimension to 64.

**SimCLR** (Tang et al., 2020) is an adaptation of the visual domain's SimCLR (Chen et al., 2020) to time series analysis. This method applies two augmentations (e.g., rotation, negation) to a time series instance, aiming to maximize the agreement between latent representations from the same instances and minimize it for different ones. Instead of using the official Tensorflow-based implementation, we developed a custom version using a ResNet-1D encoder (Wang et al., 2017). This encoder includes three blocks consisting of Conv1D, BatchNorm, and Maxpool1D operations, with filter sizes set to [32, 64, 128]. An average pooling layer on the final block generates a 128-dimensional latent representation for each instance.

**TS2Vec** (Yue et al., 2022) is a task-agnostic time series representation model widely recognized as a competitive baseline in numerous self-supervised studies. The model begins by randomly cropping time series to extract overlapping window segments. These segments are then projected using a linear layer, followed by random masking of the latent representations. Subsequently, both temporal and instance-wise contrastive loss is employed, where the loss is subsequently employed in a hierarchcial manner. Following the work of (Zhang et al., 2022), we employ two layers of TS2Vec modules, with hidden dimension of size 64 and output dimension of 128.

**TS-TCC** (Eldele et al., 2021) While we have outlined the difference between PPT and TS-TCC in Sec. 2, we provide further details of the TS-TCC implemention. TS-TCC begins by applying weak and strong augmentations to the input time series, generating two distinct views for the raw time series.

These views are then processed by the encoder layer, which produces latent representations of the time series. These representations are treated as sequences, and a context vector is constructed using a transformer model. Temporal and contextual contrasting losses are applied using this context vector. We implemented TS-TCC using the official source from (https://github.com/emadeldeen24/TS-TCC), with the following hyperparameters: kernel size=8, stride=1, timestep=10, and hidden dimensions of 64. Additionally, we used a jitter ratio of 0.8 and a jitter scale ratio of 1.1.

**SimMTM** (Dong et al., 2024) is a self-supervised pretraining framework that employs a multiple masked modeling strategy. Instead of creating a single masked series, the framework utilizes several masked versions of the time series for a reconstruction pretext task. The primary objective of this paper was to introduce a pretraining framework, with the work primarily evaluated in transfer learning scenarios. We utilized the official implementation available at (https://github.com/thuml/SimMTM), setting the hyperparameters as follows: kernel size 10, stride 5, masking ratio 0.3, masked numbers 3, and final output representation channels of 128. A significant drawback, however, is the substantial increase in computational demand proportional to the number of masked samples. As such, we tested only on three masked samples. The CNN output channels were manually adjusted based on the task, with settings of 6 for Gilon, 10 for PTB, 7 for MS HAR, and EMOPain.

**TimeMAE** (Cheng et al., 2023) is a patch-based self-supervised learning model for time series data, inspired by the masked autoencoder (MAE) approach in computer vision. The model utilizes transformer architectures with two pretext tasks: masked codeword classification and masked reconstruction of patches. We set the hidden dimension of the model to 64. The patch length is chosen task-specifically, ensuring the number of patches falls within the range of 15 to 40 for optimal performance.

**TS-GAC** (Wang et al., 2023) is an extension of TS-TCC model to graph neural network (GNN), utilizing patch representations for graph construction. The graphs are constructed by using patches as node features and patch correlations as edges, where they proposes to use a node and edge augmentation method for contrastive learning. Multiple graphs are formed within an instance, and these graph representations are considered as sequences. These sequences are then summarized using an autoregressive model (*i.e.* transformer) to construct a context vector for temporal contrasting. Additionally, the model employs wavelet-based augmentation, differentiating the intensity between weak and strong augmentations. We implemented TS-GAC using the official code from (https://github.com/Frank-Wang-oss/TS-GAC), setting the patch length to 5 for Gilon and 10 for PTB, EMO, and MS HAR, with a kernel size of 3 and stride of 1.

# I  TIME SERIES CLUSTERING

We performed time series clustering task with the learned time series representation from self-supervised linear probing as in the works of (Zhang et al., 2022; Wang et al., 2024b). We fitted K-means clustering on top of the pre-trained representations of the GL HAR task, which consists of 7 classes. The experiments were performed with 5 different random seeds to ensure robustness. We report the Silhouette Score, Normalized Mutual Information (NMI), and Adjusted Rand Index (ARI) scores to evaluate the clustering performance. We also compared our works to SOTA works such as SimCLR, TS2Vec, TF-C, TimeMAE, UniTS, and other pretext task based methods such as mask and reconstruct, and contrastive learning based method.

| Model | Silhouette Score | NMI | ARI |
|---|---|---|---|
| SimCLR | $0.128_{\pm 0.037}$ | $0.472_{\pm 0.025}$ | $0.289_{\pm 0.021}$ |
| TS2Vec | $0.128_{\pm 0.037}$ | $0.607_{\pm 0.029}$ | $0.405_{\pm 0.024}$ |
| TF-C | $0.143_{\pm 0.020}$ | $0.585_{\pm 0.035}$ | $0.429_{\pm 0.026}$ |
| TimeMAE | $0.141_{\pm 0.023}$ | $0.556_{\pm 0.017}$ | $0.429_{\pm 0.022}$ |
| UniTS | $0.308_{\pm 0.057}$ | $0.610_{\pm 0.105}$ | $0.475_{\pm 0.119}$ |
| PatchTST (Mask) | $\mathbf{0.338}_{\pm \mathbf{0.007}}$ | $0.548_{\pm 0.016}$ | $0.384_{\pm 0.018}$ |
| PatchTST (CL) | $0.253_{\pm 0.009}$ | $0.444_{\pm 0.007}$ | $0.275_{\pm 0.021}$ |
| PatchTST (PPT) | $0.263_{\pm 0.023}$ | $\mathbf{0.644}_{\pm \mathbf{0.048}}$ | $\mathbf{0.481}_{\pm \mathbf{0.050}}$ |

**Table 10:** Comparison of clustering metrics in GL task

In Tab. 10, we show that PPT achieves the best performance in NMI and ARI scores, demonstrating its effectiveness in generating meaningful representations in clustering tasks. Here, NMI and ARI measure the agreement between the predicted clusters and the ground truth labels. A higher NMI and ARI score suggests that the clustering assignment is close to the ground truth. While we acknowledge that the mask-based pretext task used in PatchTST outperforms PPT in terms of the Silhouette Score, it is important to note that the Silhouette Score is an internal evaluation metric that measures the compactness of clusters without requiring ground truth labels. A higher Silhouette Score indicates that the clusters are well-separated and compact, but it does not necessarily imply that the clustering results align with the true class labels.

The result that PPT outperforms other SOTA methods in terms of NMI and ARI scores highlights its ability to capture the internal temporal structure and order relationships within the time series, leading to clustering results that align well with the true class labels. In summary, we show that PPT demonstrates its effectiveness in generating meaningful and informative representations for clustering tasks.

## J MORE LINEAR-PROBING RESULTS

**Table 11: Linear-Probing Results**. We compare our methods to several self-supervised methods. The asterisk (*) denotes patch based methods. The Best and second best scores are in red and **black bold**.

| Dataset | Models | Accuracy | AUROC | AUPRC | F1 score | Precision | Recall |
|---|---|---|---|---|---|---|---|
| EMO | Mixing-up | 74.48±2.93 | 71.80±7.11 | 52.22±2.71 | 49.30±2.53 | 51.66±1.81 | 49.36±4.01 |
| | SimCLR | 74.42±4.38 | 72.71±7.29 | 48.34±6.76 | 44.64±6.34 | 47.83±7.07 | 47.19±8.02 |
| | TS2Vec | 78.08±2.93 | 78.00±3.58 | 50.95±3.83 | 49.07±3.00 | 48.97±3.52 | 49.87±2.88 |
| | TF-C | 77.63±6.18 | 82.22±5.31 | 54.84±7.18 | **53.30±6.90** | 56.14±8.79 | **56.00±9.07** |
| | TS-TCC | 75.61±3.49 | 73.82±7.40 | 53.98±3.77 | 48.47±3.59 | 54.75±0.67 | 49.02±4.05 |
| | SimMTM | **81.75±3.33** | 81.35±7.69 | **58.70±4.03** | 53.08±3.68 | 57.75±4.84 | 51.62±4.17 |
| | TimeMAE* | 73.97±2.28 | 70.11±4.43 | 43.25±2.14 | 42.44±2.07 | 42.81±2.37 | 42.43±2.09 |
| | TS-GAC* | 73.75±1.66 | 75.92±2.30 | 49.29±0.62 | 46.42±1.29 | 46.04±0.87 | 48.86±1.69 |
| | PatchTST(Mask)* | 78.70±0.73 | **82.60±1.39** | 55.23±2.21 | 45.81±2.07 | **59.40±5.32** | 46.35±1.31 |
| | PITS (Mask)* | 68.00±0.81 | 63.61±1.20 | 43.69±0.59 | 43.84±0.87 | 44.77±0.59 | 46.83±1.50 |
| | PatchTST (Ours)* | **81.92±0.58** | **84.74±1.55** | **62.51±3.09** | **54.19±2.33** | **62.96±2.49** | **53.41±2.42** |
| | PITS (Ours)* | 75.55±2.84 | 68.63±3.30 | 45.59±2.28 | 45.75±2.43 | 45.05±2.65 | 47.53±2.06 |
| MS HAR | Mixing-up | 83.89±1.78 | 97.59±0.62 | 84.60±3.14 | 78.99±2.30 | 81.16±2.67 | 79.00±2.17 |
| | SimCLR | 83.86±0.92 | 97.87±0.25 | 85.02±1.28 | 78.58±1.38 | 80.98±0.88 | 78.34±1.11 |
| | TS2Vec | 82.76±0.57 | 96.86±0.32 | 83.10±1.41 | 77.72±0.67 | 80.24±0.84 | 77.30±0.92 |
| | TF-C | 85.02±2.71 | 98.10±0.24 | 86.10±1.73 | 80.34±3.59 | 82.51±2.93 | 80.06±3.50 |
| | TS-TCC | 86.67±2.62 | 98.61±0.30 | 88.24±1.74 | 83.48±2.95 | 85.56±2.42 | 82.88±3.12 |
| | SimMTM | 75.07±1.13 | 95.96±0.07 | 72.55±0.95 | 67.76±1.61 | 70.51±1.64 | 67.75±1.59 |
| | TimeMAE* | 80.03±0.42 | 96.04±0.88 | 80.20±1.36 | 75.06±1.03 | 76.58±1.70 | 75.62±0.81 |
| | TS-GAC* | **88.81±1.10** | **99.01±0.21** | **92.88±0.81** | **86.12±1.37** | **88.03±1.11** | **85.34±1.47** |
| | PatchTST(Mask)* | 69.96±3.77 | 93.11±1.42 | 64.04±5.83 | 60.47±5.41 | 65.35±4.30 | 60.94±5.16 |
| | PITS (Mask)* | 84.89±1.66 | 98.21±0.22 | 88.08±1.33 | 80.95±2.26 | 83.71±2.18 | 80.43±2.21 |
| | PatchTST (Ours)* | 84.96±2.08 | 98.01±0.25 | 86.76±2.08 | 80.71±2.86 | 82.66±2.83 | 80.43±2.39 |
| | PITS (Ours)* | **89.33±0.86** | **98.58±0.38** | **91.09±1.00** | **86.54±1.05** | **88.76±1.06** | **85.70±1.07** |
| PTB | Mixing-up | 82.35±0.93 | 89.40±2.22 | 95.32±1.05 | 88.82±0.57 | 81.76±0.72 | 97.23±0.61 |
| | SimCLR | 80.41±1.14 | 86.03±3.24 | 93.84±1.71 | 87.89±0.62 | 79.30±0.99 | 98.57±0.21 |
| | TS2Vec | **84.06±2.81** | **91.56±3.28** | **96.17±2.02** | **89.75±1.68** | **83.82±2.56** | 96.62±0.95 |
| | TF-C | 79.58±1.22 | 86.55±2.64 | 93.95±1.50 | 87.25±0.85 | 79.35±0.47 | 96.91±1.78 |
| | TS-TCC | 79.58±1.18 | 79.18±1.88 | 88.05±1.40 | 86.98±0.78 | 80.51±0.90 | 94.59±1.47 |
| | SimMTM | 80.75±0.29 | 88.66±1.96 | 94.48±1.23 | 88.08±0.17 | 79.58±0.31 | **98.61±0.41** |
| | TimeMAE* | 78.65±1.43 | 84.72±1.30 | 92.97±0.83 | 86.78±0.86 | 78.41±1.03 | 97.15±1.10 |
| | TS-GAC* | 83.26±1.97 | 91.18±2.36 | 95.37±0.96 | 89.51±1.09 | 81.76±1.95 | **98.90±0.23** |
| | PatchTST(Mask)* | 79.89±1.87 | 81.36±4.91 | 88.94±3.50 | 87.46±1.07 | 79.50±1.54 | 97.21±1.07 |
| | PITS (Mask)* | 81.48±1.50 | 81.98±3.18 | 90.77±1.79 | 88.22±0.87 | 81.53±1.40 | 96.13±0.82 |
| | PatchTST (Ours)* | 78.50±1.76 | 76.04±6.47 | 86.79±5.56 | 88.67±0.95 | 78.45±1.62 | 96.83±0.63 |
| | PITS (Ours)* | **86.48±0.40** | **91.83±1.36** | **96.26±0.85** | **91.24±0.26** | **85.67±0.61** | 97.58±0.86 |
| GL | Mixing-up | 81.76±2.94 | 96.46±0.50 | 82.43±2.80 | 79.25±3.85 | 79.99±3.83 | 79.18±3.93 |
| | SimCLR | 87.10±0.44 | 98.25±0.12 | 92.69±1.04 | 88.72±0.79 | 89.42±0.84 | 88.20±0.84 |
| | TS2Vec | 87.79±0.66 | 98.35±0.12 | 93.93±0.10 | 89.92±0.68 | 90.12±0.97 | 89.86±0.42 |
| | TF-C | 88.23±0.73 | 98.66±0.13 | 92.90±1.00 | 89.09±1.11 | 89.30±1.12 | 89.00±1.02 |
| | TS-TCC | 87.98±1.28 | 98.60±0.22 | 91.86±1.03 | 88.53±1.32 | 88.82±1.45 | 88.36±1.27 |
| | SimMTM | 82.05±0.33 | 97.08±0.06 | 88.06±0.49 | 83.96±0.50 | 84.63±0.64 | 83.63±0.47 |
| | TimeMAE* | 83.44±1.09 | 97.81±0.11 | 89.84±1.09 | 85.26±1.74 | 86.10±2.06 | 84.77±1.58 |
| | TS-GAC* | **92.32±0.85** | 99.04±0.38 | 96.23±0.64 | **92.79±0.84** | 92.67±0.65 | **93.23±0.92** |
| | PatchTST(Mask)* | 88.43±0.44 | 98.79±0.11 | 94.34±0.40 | 89.87±0.38 | 89.73±0.45 | 90.15±0.43 |
| | PITS (Mask)* | 89.29±0.76 | 98.79±0.11 | 93.62±0.45 | 89.04±0.40 | 89.72±0.50 | 88.75±0.38 |
| | PatchTST (Ours)* | **92.33±0.48** | **99.28±0.10** | **96.83±0.44** | **93.67±0.45** | **93.54±0.37** | **93.90±0.56** |
| | PITS (Ours)* | 92.07±0.78 | **99.28±0.09** | **96.87±0.46** | 92.77±0.63 | **92.99±0.70** | 92.83±0.65 |

# K MORE FINE-TUNING RESULTS

**Table 12: Full fine-tuning results of Gilon**. We first perform self-supervised pretraining, followed by full fine-tuning the whole network with 100%, 10%, and 1% of training data in Gilon task. Best scores for PatchTST and PITS are in **red bold** and the top scoring value is underscored.

| Fraction | Models | Accuracy | AUROC | AUPRC | F1 score | Precision | Recall |
|---|---|---|---|---|---|---|---|
| 100% | Mixing-up | 93.92±0.85 | 98.85±0.19 | 95.18±1.07 | 91.71±0.98 | 91.55±1.03 | 92.11±0.85 |
| | SimCLR | 95.78±0.36 | 99.64±0.03 | 98.52±0.13 | 95.47±0.38 | 95.66±0.51 | 95.40±0.32 |
| | TS2Vec | 91.77±0.75 | 98.17±0.51 | 94.04±1.80 | 90.28±1.26 | 90.52±1.29 | 90.17±1.25 |
| | TF-C | 91.98±0.62 | 99.23±0.08 | 95.60±0.59 | 91.45±0.71 | 91.32±0.73 | 91.77±0.72 |
| | TS-TCC | 94.58±0.70 | 99.39±0.08 | 97.38±0.30 | 92.61±1.00 | 93.39±0.92 | 92.61±0.91 |
| | SimMTM | 94.83±1.04 | 98.99±0.48 | 96.27±1.33 | 93.44±1.29 | 93.62±1.28 | 93.57±1.04 |
| | TimeMAE | 95.63±0.31 | 99.43±0.15 | 97.60±0.56 | 94.61±0.55 | 94.71±0.45 | 94.60±0.68 |
| | TS-GAC | 96.43±0.52 | 99.71±0.08 | 98.88±0.35 | 96.20±0.55 | 96.35±0.69 | 96.19±0.42 |
| | PatchTST (Random Init) | 95.01±4.20 | 99.61±0.09 | 98.58±0.40 | 95.52±2.43 | 96.15±1.90 | 95.60±2.61 |
| | PatchTST (Mask) | **97.12±0.49** | 99.73±0.04 | 99.00±0.14 | **96.94±0.57** | **97.30±0.35** | **97.09±0.47** |
| | PatchTST (Ours) | 96.71±0.86 | **99.76±0.04** | **99.12±0.10** | 96.70±0.71 | 97.03±0.37 | 96.80±0.50 |
| | PITS (Random Init) | 93.40±0.34 | 99.54±0.11 | 97.58±0.60 | 92.96±0.59 | 92.76±0.67 | 92.73±0.60 |
| | PITS (Mask + CL) | 91.44±1.84 | 99.15±0.19 | 95.06±1.27 | 90.93±1.35 | 90.83±1.48 | 90.73±1.51 |
| | PITS (Ours) | **95.53±0.83** | **99.79±0.06** | **98.89±0.37** | **95.25±0.84** | **95.43±0.89** | **95.25±0.76** |
| 10% | Mixing-up | 92.85±0.69 | 98.74±0.26 | 94.42±1.14 | 90.44±0.91 | 90.59±0.65 | 90.69±0.90 |
| | SimCLR | 84.55±0.78 | 98.47±0.20 | 91.74±0.83 | 83.60±1.21 | 86.78±0.62 | 82.56±1.16 |
| | TS2Vec | 88.12±1.58 | 96.46±0.78 | 89.37±1.87 | 85.51±1.13 | 85.86±0.69 | 85.97±1.71 |
| | TF-C | 83.35±0.48 | 97.96±0.09 | 87.87±0.36 | 82.73±0.48 | 83.95±0.33 | 82.10±0.59 |
| | TS-TCC | 93.69±1.05 | 99.41±0.19 | 97.36±0.76 | 92.11±0.84 | 93.69±0.38 | 91.83±0.83 |
| | SimMTM | 91.94±0.58 | 98.95±0.35 | 95.65±0.62 | 91.35±0.53 | 91.41±0.44 | 91.40±0.65 |
| | TimeMAE | 90.06±2.95 | 98.77±0.43 | 94.65±2.04 | 91.10±2.54 | 91.50±2.46 | 90.83±2.63 |
| | TS-GAC | 94.51±1.73 | 99.50±0.05 | 97.98±0.35 | 94.20±1.69 | 94.47±1.73 | 94.25±1.41 |
| | PatchTST (Random Init) | 88.92±3.54 | 99.25±0.09 | 96.43±0.59 | 90.44±2.34 | 91.68±1.51 | 90.55±2.38 |
| | PatchTST (Mask) | 91.61±0.82 | 99.35±0.11 | 97.10±0.47 | 92.33±0.89 | 92.88±0.80 | 92.47±0.75 |
| | PatchTST (Ours) | **93.26±1.57** | **99.50±0.09** | **97.79±0.47** | **93.97±1.40** | **94.74±1.23** | **94.27±1.34** |
| | PITS (Random Init) | 87.91±2.14 | 98.58±0.23 | 91.47±1.97 | 88.36±2.19 | 87.22±1.78 | 87.58±1.94 |
| | PITS (Mask + CL) | 85.11±3.78 | 98.18±0.43 | 89.51±2.63 | 85.67±2.21 | 84.61±2.35 | 84.60±2.65 |
| | PITS (Ours) | **92.47±1.06** | **99.48±0.12** | **97.28±0.78** | **93.32±0.60** | **93.17±0.69** | **93.07±0.46** |
| 1% | Mixing-up | 84.82±2.17 | 97.27±0.53 | 87.48±1.81 | 82.08±2.85 | 83.76±2.52 | 81.53±3.34 |
| | SimCLR | 62.61±1.89 | 90.88±2.03 | 66.05±4.28 | 47.28±4.56 | 63.15±9.38 | 51.63±2.92 |
| | TS2Vec | 77.41±1.33 | 96.17±0.45 | 82.84±1.67 | 75.17±2.85 | 79.04±1.10 | 74.64±3.01 |
| | TF-C | 65.34±2.50 | 91.19±1.59 | 71.15±3.38 | 52.88±4.98 | 71.92±4.11 | 52.95±3.65 |
| | TS-TCC | 85.77±1.08 | 97.82±0.25 | 89.85±1.19 | 83.02±1.16 | 86.31±2.00 | 83.04±1.46 |
| | SimMTM | 78.44±2.20 | 94.93±0.87 | 82.75±1.34 | 79.48±1.95 | 80.66±2.40 | 79.31±1.85 |
| | TimeMAE | 76.09±2.01 | 96.24±0.53 | 80.35±3.35 | 74.63±3.30 | 77.58±3.69 | 73.57±3.61 |
| | TS-GAC | 80.57±11.8 | 97.28±1.67 | 87.57±6.06 | 72.99±19.7 | 82.01±7.92 | 74.02±17.5 |
| | PatchTST (Random Init) | 82.63±2.89 | 97.89±0.74 | 89.84±2.02 | 85.16±2.35 | 82.78±4.48 | 83.07±3.76 |
| | PatchTST (Mask) | 80.55±2.29 | 96.77±1.04 | 86.44±2.83 | 83.26±2.11 | 81.50±3.58 | 81.52±3.58 |
| | PatchTST (Ours) | **84.80±1.68** | **98.08±0.38** | **90.64±1.90** | **86.92±1.48** | **86.88±1.65** | **86.75±1.57** |
| | PITS (Random Init) | 74.25±3.03 | 96.15±0.61 | 78.73±2.90 | 75.27±4.24 | 71.45±4.74 | 72.08±5.65 |
| | PITS (Mask + CL) | 72.41±2.05 | 95.40±0.60 | 75.92±3.23 | 72.81±4.76 | 69.83±3.77 | 70.45±4.71 |
| | PITS (Ours) | **81.04±1.86** | **97.68±0.30** | **87.26±1.62** | **83.71±0.95** | **81.25±1.45** | **82.05±1.30** |

# L  ACF-CoS

## L.1  PURPOSE OF DESIGNING ACF-CoS

An autocorrelation function (ACF) calculates the correlation coefficients between a time series and its own lagged versions at various intervals. Here, ACF provides an overview of the structured linear patterns within the time series, revealing specific lags where these patterns recur. For example, when applied to a time series from a force sensor in a smart insole, the ACF shows regular peaks that may correspond to step frequency, indirectly providing information regarding the user's pace. As such, ACF provides a meaningful overview of a time series's structural dependency and order patterns.

Accordingly, we designed the ACF-CoS metric, which measures the cosine similarity between the ACF of a time series and its patch-permuted version to quantify the order information present. To our knowledge, no existing metric systematically quantifies the amount of order information in a time series. The closest relevant metric, the permutation entropy (PE) (Bandt & Pompe, 2002), measures the frequencies of a particular symbolic pattern to occur within the series. This diverges from our objective of measuring "orderness" by focusing instead on pattern frequency rather than the structural order itself. The intuition behind ACF-CoS is that the ACF of a time series with "orderness" should strongly diverge from the ACF of its patch-permuted version, leading to smaller ACF-CoS scores. On the other hand, time series with no order information (i.e., white noise) should have a similar ACF score for its patch-permuted version, leading to high ACF-CoS scores.

**Discussion on Weak Stationary Assumption of ACF-CoS.** ACF has been primarily used in the time series forecasting domain, where they analyze the absolute values of the ACF to understand trend and seasonal patterns across different time windows. Holding the weak stationary assumption (having constant mean and variance) is important for autocorrelation estimates in such scenarios. The assumption enables the comparison of the ACF against different time steps or datasets. Unlike such comparisons made with varying steps of time or datasets, our ACF-CoS does not rely on absolute values of ACF but on the similarity between patterns. Even if the data are non-stationary, our primary objective is to quantify how autocorrelation structure changes when the patches' order is shuffled. As such, we are less concerned with the stationary assumption (which may be affected when we deal with the absolute values) in calculating ACF-CoS.

## L.2  FULL ACF-CoS SCORES

We provide the detailed ACF-CoS statistics in Tab. 13. The ACF score, calculated for 17 tasks from the UEA Repository, is the average score across three different patch sizes  (Tab. 14). The ACF Range provides the minimum and maximum value of ACF-CoS score. $T$ and $D$ is the length of the time sequence and number of channels, respectively. The AUROC Win (%) metric shows how often our method (PPT) surpassed the cross-entropy (CE) based method by at least 1% in the AUROC metric. Similarly, the Accuracy Win (%) measures how frequently Method outperformed the CE method in accuracy. A total of 162 different hyperparameter configurations were tested for each dataset.

## L.3  HYPERPARAMETER CONFIGURATIONS FOR ACF-CoS CALCULATION

We calculated the ACF-CoS score based on the hyperparameter configuration in Tab. 14. We selected a patch size allowing between 10 to 40 patches per dataset. The batch size was determined by the number of training samples. The hidden dim is the size of hidden dimension in PatchTST. Strong $\gamma$ is the lower bound of strong permutation frequency used in PPT training. We used two different learning rates throughout the whole training: 0.02 and 0.002. For the EigenWorms dataset, data sampling was necessary due to its large size. The total combination of hyperparameters are 162.

**Table 13:** ACF-CoS Statistics

| Dataset Name | ACF-CoS | ACF Range | $T$ | $D$ | AUROC Win (%) | Acc. Win (%) | # Acc. Wins |
|---|---|---|---|---|---|---|---|
| ShapesAll | 0.449 | 0.239 - 0.660 | 512 | 1 | 44.44 | 62.96 | 102 / 162 |
| UWaveGestureLibraryAll | 0.423 | 0.262 - 0.585 | 945 | 1 | 35.19 | 46.30 | 75 / 162 |
| Cricket | 0.418 | 0.261 - 0.576 | 1197 | 6 | 61.73 | 75.93 | 123 / 162 |
| OSULeaf | 0.418 | 0.266 - 0.570 | 427 | 1 | 37.04 | 47.53 | 77 / 162 |
| CricketX | 0.416 | 0.280 - 0.552 | 300 | 1 | 27.16 | 46.91 | 76 / 162 |
| CharacterTrajectories | 0.410 | 0.207 - 0.613 | 120 | 3 | 44.44 | 53.09 | 86 / 162 |
| CricketZ | 0.400 | 0.253 - 0.547 | 300 | 1 | 25.31 | 45.06 | 73 / 162 |
| CricketY | 0.397 | 0.272 - 0.522 | 300 | 1 | 48.77 | 57.41 | 93 / 162 |
| Plane | 0.374 | 0.175 - 0.574 | 144 | 1 | 37.04 | 43.83 | 71 / 162 |
| Handwriting | 0.335 | 0.239 - 0.431 | 152 | 3 | 46.30 | 51.85 | 84 / 162 |
| EigenWorms | 0.290 | 0.216 - 0.363 | 17984 | 6 | 58.02 | 61.11 | 99 / 162 |
| Earthquakes | 0.262 | 0.209 - 0.316 | 512 | 1 | 43.21 | 42.59 | 69 / 162 |
| Blink | 0.223 | 0.132 - 0.313 | 510 | 4 | 45.06 | 43.21 | 70 / 162 |
| Libras | 0.218 | 0.091 - 0.345 | 45 | 2 | 30.25 | 46.91 | 76 / 162 |
| NATOPS | 0.216 | 0.087 - 0.346 | 51 | 24 | 53.70 | 75.31 | 122 / 162 |
| LargeKitchenAppliances | 0.190 | 0.151 - 0.230 | 720 | 1 | 48.77 | 54.32 | 88 / 162 |
| GestureMidAirD1 | 0.186 | 0.104 - 0.268 | 360 | 1 | 40.74 | 48.15 | 78 / 162 |
| WalkingSittingStanding | 0.176 | 0.086 - 0.267 | 206 | 3 | 43.21 | 46.30 | 75 / 162 |
| GestureMidAirD2 | 0.166 | 0.099 - 0.234 | 360 | 1 | 37.04 | 46.30 | 75 / 162 |
| GunPointOldVersusYoung | 0.105 | 0.028 - 0.182 | 150 | 1 | 3.09 | 7.41 | 12 / 162 |
| GunPointAgeSpan | 0.104 | 0.024 - 0.183 | 150 | 1 | 45.68 | 46.91 | 76 / 162 |
| GunPointMaleVersusFemale | 0.100 | 0.029 - 0.171 | 150 | 1 | 28.40 | 45.68 | 74 / 162 |
| GestureMidAirD3 | 0.060 | 0.034 - 0.086 | 360 | 1 | 27.16 | 37.65 | 61 / 162 |
| EMOPain | 0.026 | 0.018 - 0.035 | 200 | 30 | 40.74 | 33.33 | 54 / 162 |

**Table 14:** Hyperparameter Search Space for ACF-CoS Calculation.

| Dataset Name | Patch Size | Batch Size | Hidden Dim | Strong $\gamma$ | learning_rate | Use All Data? |
|---|---|---|---|---|---|---|
| ShapesAll | 13, 17, 26 | 4, 8, 16 | 16, 32, 64 | 10, 20, 40 | 0.02, 0.002 | True |
| UWaveGestureLibraryAll | 24, 32, 47 | 32, 64, 128 | 16, 32, 64 | 10, 20, 40 | 0.02, 0.002 | True |
| Cricket | 30, 40, 50 | 4, 8, 16 | 16, 32, 64 | 10, 20, 40 | 0.02, 0.002 | True |
| OSULeaf | 11, 14, 21 | 4, 8, 16 | 16, 32, 64 | 10, 20, 40 | 0.02, 0.002 | True |
| CricketX | 8, 10, 15 | 4, 8, 16 | 16, 32, 64 | 10, 20, 40 | 0.02, 0.002 | True |
| CharacterTrajectories | 3, 4, 6 | 32, 64, 128 | 32, 64, 128 | 10, 20, 40 | 0.02, 0.002 | True |
| CricketZ | 8, 10, 15 | 4, 8, 16 | 16, 32, 64 | 10, 20, 40 | 0.02, 0.002 | True |
| CricketY | 8, 10, 15 | 4, 8, 16 | 16, 32, 64 | 10, 20, 40 | 0.02, 0.002 | True |
| Plane | 3, 4, 7 | 4, 8, 16 | 16, 32, 64 | 10, 20, 40 | 0.02, 0.002 | True |
| Handwriting | 4, 5, 8 | 4, 8, 16 | 16, 32, 64 | 10, 20, 40 | 0.02, 0.002 | True |
| EigenWorms | 450, 600, 900 | 4, 8, 16 | 16, 32, 64 | 10, 20, 40 | 0.02, 0.002 | False (3000 samples) |
| Earthquakes | 13, 17, 26 | 4, 8, 16 | 16, 32, 64 | 10, 20, 40 | 0.02, 0.002 | True |
| Blink | 13, 17, 26 | 4, 8, 16 | 16, 32, 64 | 10, 20, 40 | 0.02, 0.002 | True |
| Libras | 1, 2, 4 | 4, 8, 16 | 16, 32, 64 | 10, 20, 40 | 0.02, 0.002 | True |
| NATOPS | 1, 2, 4 | 4, 8, 16 | 16, 32, 64 | 10, 20, 40 | 0.02, 0.002 | True |
| LargeKitchenAppliances | 18, 24, 36 | 32, 64, 128 | 16, 32, 64 | 10, 20, 40 | 0.02, 0.002 | True |
| GestureMidAirD1 | 9, 12, 18 | 4, 8, 16 | 16, 32, 64 | 10, 20, 40 | 0.02, 0.002 | True |
| WalkingSittingStanding | 5, 7, 10 | 32, 64, 128 | 16, 32, 64 | 10, 20, 40 | 0.02, 0.002 | True |
| GestureMidAirD2 | 9, 12, 18 | 4, 8, 16 | 16, 32, 64 | 10, 20, 40 | 0.02, 0.002 | True |
| GunPointOldVersusYoung | 3, 5, 7 | 4, 8, 16 | 16, 32, 64 | 10, 20, 40 | 0.02, 0.002 | True |
| GunPointAgeSpan | 3, 5, 7 | 4, 8, 16 | 16, 32, 64 | 10, 20, 40 | 0.02, 0.002 | True |
| GunPointMaleVersusFemale | 3, 5, 7 | 4, 8, 16 | 16, 32, 64 | 10, 20, 40 | 0.02, 0.002 | True |
| GestureMidAirD3 | 9, 12, 18 | 4, 8, 16 | 16, 32, 64 | 10, 20, 40 | 0.02, 0.002 | True |
| EMOPain | 5, 7, 10 | 32, 64, 128 | 16, 32, 64 | 10, 20, 40 | 0.02, 0.002 | True |

## L.4 STATISTICAL RESULTS

We provide the statistical results of fitting the ordinary least square (OLS) model and the robust linear regression (RLM) model on the correlation between ACF-COS score and Accuracy Win (%).

**Table 15:** OLS Regression Results

| Model: | OLS Regression | R-squared: | 0.225 |
|---|---|---|---|
| Method: | Least Squares | Adj. R-squared: | 0.189 |
| No. Observations: | 24 | F-statistic: | 6.370 |
| Df Residuals: | 22 | Prob (F-statistic): | 0.0193 |
| Df Model: | 1 | Log-Likelihood: | 17.692 |
| AIC: | -31.38 | BIC: | -29.03 |
| Omnibus: | 5.298 | Prob(Omnibus): | 0.071 |
| Skew: | -0.186 | Kurtosis: | 5.142 |
| Durbin-Watson: | 2.435 | Cond. No.: | 8.14 |

| | coef | std err | t | P> \|t\| | [0.025 | 0.975] |
|---|---|---|---|---|---|---|
| const | 0.3604 | 0.055 | 6.495 | 0.000 | 0.245 | 0.476 |
| x1 | 0.4730 | 0.187 | 2.524 | 0.019 | 0.084 | 0.862 |

**Table 16:** Robust Linear Model Regression Results

| Dep. Variable: | y | No. Observations: | 24 |
|---|---|---|---|
| Model: | RLM | Df Residuals: | 22 |
| Method: | IRLS | Df Model: | 1 |
| Norm: | HuberT | Scale Est.: | mad |
| Cov Type: | H1 | No. Iterations: | 11 |

| | coef | std err | z | P> \|z\| | [0.025 | 0.975] |
|---|---|---|---|---|---|---|
| const | 0.3870 | 0.039 | 10.022 | 0.000 | 0.311 | 0.463 |
| x1 | 0.3676 | 0.130 | 2.819 | 0.005 | 0.112 | 0.623 |

## M  TIME SERIES FOUNDATION MODEL: GPT4TS

We applied PPT to GPT4TS (Zhou et al., 2024), a time series foundation model that fine-tunes specific layers of the GPT2 (Radford et al., 2019) architecture. GPT4TS employs patching on the original time series sequences. However, we encountered limitations in fully exploiting PPT's capabilities with GPT4TS due to two primary factors: (1) GPT4TS, being based on a language model, assumes a univariate patch sequence. Consequently, in the original paper, patches were concatenated along the channel dimension for multivariate time series. This makes the use of feature-wise consistency and contrastive learning from PPT inapplicable. (2) GPT4TS utilizes overlapping patch setups on certain datasets, whereas PPT assumes non-overlapping patches. Overlapping patches can mix order information between patches, potentially reducing PPT's effectiveness. Given these constraints, we reconducted the time series classification experiments on the 10 UEA multivariate benchmark datasets using a non-overlapping patch setup, as described in the original paper. With GPT4TS, we performed a grid search using weight combinations of [(0.1, 0.1), (0.1, 0.5), (0.5, 0.1), (0.5, 0.5)] to balance the losses between the consistency and contrastive terms.

**Table 17:** Comparison of Mask and Reconstruction pretext task (each with 10 and 40% masking ratio) and PPT in self-supervised linear probing setup, using GPT4TS model.

| Dataset Name | Mask (10%) | Mask (40%) | PPT (Ours) |
|---|---|---|---|
| FaceDetection | 67.5 | 67.4 | 66.4 |
| SelfRegulationSCP1 | 91.8 | 92.5 | 93.2 |
| EthanolConcentration | 30.8 | 33.5 | 34.6 |
| Handwriting | 27.3 | 29.3 | 24.2 |
| JapaneseVowels | 97.6 | 97.6 | 98.4 |
| Heartbeat | 77.6 | 77.6 | 76.6 |
| UWaveGestureLibrary | 85.3 | 84.4 | 85.6 |
| SpokenArabicDigits | 98.5 | 98.5 | 98.1 |
| PEMS-SF | 69.9 | 76.3 | 82.1 |
| SelfRegulationSCP2 | 57.2 | 56.7 | 56.1 |
| **Average** | 70.3 | 71.4 | 71.5 |

Tab. 17 shows the results for linear probing, where we compared PPT to mask and reconstruction pretext tasks. Here, we first conducted self-supervised training with unlabeled training samples and consequently fitted a logistic regression for linear probing. We compared our work against mask and reconstruction pretext task with 10% and 40% masking ratio. We set all other hyperparameters and architectures fixed to make results comparable. We show that PPT leads to an average accuracy of 71.5% which is better or comparable to mask and reconstruct pretext task.

**Table 18:** Supervised Experiment of GPT4TS (CE) and GPT4TS (CE+PPT) Performance

| Dataset Name | GPT4TS (CE) | GPT4TS (CE+PPT) | Performance Change (%p) |
|---|---|---|---|
| FaceDetection | 67.6 | 68.2 | +0.68 |
| SelfRegulationSCP1 | 92.5 | 92.2 | -0.34 |
| EthanolConcentration | 30.0 | 32.3 | +2.28 |
| Handwriting | 29.6 | 31.1 | +1.41 |
| JapaneseVowels | 98.6 | 99.7 | +1.08 |
| Heartbeat | 75.1 | 76.1 | +0.98 |
| UWaveGestureLibrary | 85.3 | 86.9 | +1.56 |
| SpokenArabicDigits | 98.7 | 98.5 | -0.23 |
| PEMS-SF | 81.5 | 86.7 | +5.20 |
| SelfRegulationSCP2 | 54.4 | 58.9 | +4.44 |
| **Average** | 71.3 | 73.1 | +1.71 |

We also carried out a supervised experiment in which we incorporate PPT into the cross-entropy minimization process. The results in Tab. 18 show that incorporation of PPT can lead to an improvement in the performance of the average accuracy from 71.3% to 73.1%. In particular, this performance gain was achieved solely by using time-wise consistency and contrastive terms. Consequently, we anticipate further performance enhancements when PPT is integrated into time series foundation models capable of considering patches in the feature dimension, such as PatchTST and PITS.

