# OpenReview forum: "PPT: Patch Order Do Matters In Time Series Pretext Task"
_ICLR.cc/2025/Conference — ICLR 2025 Poster_

### Official Review · Reviewer_GFNC · 2024-10-27

**Soundness:** 2
**Presentation:** 2
**Contribution:** 2
**Rating:** 3
**Confidence:** 4

**Summary:**

The paper proposes a self supervised patch order learning pretext tasks for time series classification. Experiments are shown to quantify the benefits of path order learning. An evaluation metric to measure importance of orderness is also proposed.

**Strengths:**

The paper offers a perspective on ordering of patches in self supervised TS learning, and presents experiments on some datasets to show benefits and drawbacks of the approach.

**Weaknesses:**

1. Lack of SoTA baselines, compare against and cite:

- S. Gao et al, UNITS: A Unified Multi-Task Time Series Model, https://arxiv.org/abs/2403.00131, NeurIPS 2024
- Yong Liu et al, itransformer: Inverted transformers are effective for time series forecasting, ICLR 2024
- Shizhan Liu et al. Pyraformer: Low-complexity pyramidal attention for long-range time series modeling and forecasting, ICLR 2021
- Haixu Wu et al. Autoformer: Decomposition transformers with auto-correlation for long-term series forecasting, NeurIPS 2021

In particular, UniTS has been shown to outperform PatchTST and other baselines in many setups.

2. The authors do not consider tasks beyond classification. I'm not convinced that the method is truly better than others when considering the representations generated, for example no comparisons with SoTA is shown on anomaly detection or clustering tasks. See for example, TF-C Appendix G, which is a baseline method authors compare against.

- Zhang et al, Self-Supervised Contrastive Pre-Training for Time Series via Time-Frequency Consistency, NeurIPS 2022

3. Limited evaluations. Experiments are all one-to-one dataset. The authors do not consider one-to-many, or other settings to test whether the representations are generalizable to cross-dataset settings etc.

4. The gains are not convincing since many other baselines outperform the proposed method in several metrics.The improvement over GPT4TS is marginal.

5. It's not clear whether this method even works for diverse dataset pretraining settings as in UniTS (https://arxiv.org/abs/2403.00131). Since this method is mainly for pretext tasks, this is an important point to raise given we're in the era of foundation models.

**Questions:**

See weaknesses.

---

> ### Author Response · Authors · 2024-11-19
>
> We thank the reviewer for providing constructive feedback and insights to improve our work. As the reviewer mentioned, we have thoroughly detailed the strengths and weaknesses of our order-based supervisions for time series throughout our paper. We have also proposed a novel metric ACF-COS, which helps us overcome our weaknesses by pre-assessing the order-awareness in time series tasks. In response to the reviewer's feedback, we have made the following improvements in our work:
>
> 1. We conducted extensive baseline self-supervised comparisons with UniTS on three different datasets, showing that PPT outperforms UniTS.
>
> 2. We conducted additional time series clustering task, where we show that PPT outperforms in NMI, and ARI metrics where we compare the clustering performance with respect to the ground truths. Accordingly, we have updated our manuscript in **Appendix I. Time Series Clustering** to incorporate our results.
>
> Here, we have prepared a detailed response to the reviewer's questions. We carefully address each of the comments below.

---

> > ### Author Response · Authors · 2024-11-19
> >
> > **W1. Lack of SoTA baselines, cite UniTS, iTransformer, Pyraformer, Autoformer.**
> >
> > A1-1. We thank the reviewer for bringing our attention to works that are relevant to ours and helping us to strengthen our related works. Our response is twofold.
> >
> > First, we respectfully disagree that our work lacks SoTA baselines. We have made extensive comparisons against the most recent and relevant SSL SoTA works for time series classification, such as PITS (ICLR 2024), TS-GAC (AAAI 2024), SimMTM (NeurIPS 2023), PatchTST (ICLR 2023), and Tf-C (NeurIPS 2022). We have also shown that our pretext task can be incorporated into both linear-based (PITS) and transformer-based (PatchTST) SoTA models, showcasing their effectiveness in various tasks (e.g., Self-Supervised, Semi-Supervised, and Supervised).
> >
> > Second, as the reviewer suggested, we have cited iTransformer and Autoformer which are the SoTA works in long-term time series forecasting (LTSF) problems (Line 520).

---

> > > ### Author Response · Authors · 2024-11-19
> > >
> > > **A1-2.** We thank the reviewer for bringing our attention to UniTS, a relevant work to PPT, which was accepted to NeurIPS 2024. We would first like to explain why we were not able to compare our work to UniTS initially and then discuss the experimental comparison with UniTS.
> > >
> > > Firstly, we were unable to compare our work to UniTS at the time of submission, as the NeurIPS 2024 acceptance date (Sep 26) and ICLR submission dates (Sep 27) were very close. However, we have cited and made extensive comparisons to UniTS in our updated manuscript. We have compared our work with UniTS on self-supervised linear probing and clustering tasks.
> > >
> > > For the hyperparameters of UniTS, we conducted a grid search on the learning rate between [5e-5, 5e-4, 1e-4, 1e-3] and d_model between [32, 64]. We set the accumulate gradients to 1 as our work focuses on a one-to-one setup. We report the results with a learning rate of 1e-4 (EMOPain, PTB) and 1e-3 (GL) and d_model=64, as they performed the best.
> > >
> > > Briefly, while UniTS shows strong and competitive performance, it did not outperform PPT in three of the benchmarks we used. This might be attributed to UniTS's novel contribution towards time series multi-task learning, which focuses on learning generalized representations from multiple tasks such as forecasting, classification, and anomaly detection. Unlike UniTS, PPT is more focused on providing order-aware supervision to time series classification tasks that exhibit order dependency across time and channels, where our focus is to provide a generalized representation within the same time series classification.
> > >
> > > As we have noted in our limitation section, PPT did not perform well for forecasting tasks, as permuting the orders of patches for forecasting disturbs the trend pattern, a key factor in time series forecasting. However, PPT is highly effective for tasks that show order information in time series classification. We intend to explore the integration of PPT with multi-task learning frameworks like UniTS to learn more generalized, task-agnostic time series representations in future work.
> > >
> > > We have also updated the main manuscript to include the comparisons with UniTS and added the following tables to showcase the results:
> > >
> > > **Linear Probing Experiment**
> > > | Dataset | Model | Accuracy | F1 score | AUROC | AUPRC |
> > > |---------|-------|----------|-----------|--------|--------|
> > > | EMO | UniTS | 78.37±0.13 | 29.51±0.51 | 65.95±2.95 | 44.84±2.39 |
> > > | EMO | PatchTST (+PPT) | **81.92±0.58** | **54.19±2.33** | **84.74±1.55** | **62.51±3.09** |
> > >
> > > | Dataset | Model | Accuracy | F1 score | AUROC | AUPRC |
> > > |---------|-------|----------|-----------|--------|--------|
> > > | PTB | UniTS* | 84.20±1.01 | 89.57±0.60 | 88.98±1.35 | 94.81±0.84 |
> > > | PTB | PITS (+PPT) | **86.48±0.40** | **91.24±0.26** | **91.83±1.36** | **96.26±0.85** |
> > >
> > > | Dataset | Model | Accuracy | F1 score | AUROC | AUPRC |
> > > |---------|-------|----------|-----------|--------|--------|
> > > | GL | UniTS* | 84.45±0.98 | 83.93±1.21 | 98.27±0.18 | 90.03±0.93 |
> > > | GL | PatchTST* (+PPT) | **92.33±0.48** | **93.67±0.45** | **99.28±0.10** | **96.83±0.44** |
> > >
> > > In summary, while UniTS demonstrates strong performance, PPT outperforms it on the time series classification datasets. We believe that the order-aware supervision provided by PPT is **particularly beneficial for time series classification tasks with order dependencies**. We have updated our manuscript to include these comparisons and insights.

---

> > > > ### Author Response · Authors · 2024-11-19
> > > >
> > > > **Q2. The authors do not consider tasks beyond classification. I'm not convinced that the method is truly better than others when considering the representations generated, for example no comparisons with SoTA is shown on anomaly detection or clustering tasks.**
> > > >
> > > > We appreciate the reviewer's feedback and suggestion to evaluate our method on tasks beyond classification. Addressing this feedback, we conducted clustering experiments, as in the works of TF-C, to assess the quality of the representations generated by our method.
> > > >
> > > > We fitted K-means clustering on top of the pre-trained representations of the GL HAR task, which consists of 7 classes. The experiments were performed with 5 different random seeds to ensure robustness. We report the Silhouette Score, Normalized Mutual Information (NMI), and Adjusted Rand Index (ARI) scores to evaluate the clustering performance. We also compared our works to SOTA works such as SimCLR, TS2Vec, TF-C, TimeMAE, UniTS, and other pretext task based methods such as mask and reconstruct, and contrastive learning-based method.
> > > >
> > > > In the below table, we show that PPT achieves the best performance in NMI and ARI scores, demonstrating its effectiveness in generating meaningful representations in clustering tasks. Here, NMI and ARI measure the agreement between the predicted clusters and the ground truth labels. A higher NMI and ARI score suggests that the clustering assignment is close to the ground truth. While we acknowledge that the mask-based pretext task used in PatchTST outperforms PPT in terms of the Silhouette Score, it is important to note that the Silhouette Score is an internal evaluation metric that measures the compactness of clusters without requiring ground truth labels. A higher Silhouette Score indicates that the clusters are well-separated and compact, but it does not necessarily imply that the clustering results align with the true class labels.
> > > >
> > > > | Model | Silhouette Score | NMI | ARI |
> > > > |-------|------------------|-----|-----|
> > > > | SimCLR | 0.128±0.037 | 0.472±0.025 | 0.289±0.021 |
> > > > | TS2Vec | 0.128±0.037 | 0.607±0.029 | 0.405±0.024 |
> > > > | TF-C | 0.143±0.020 | 0.585±0.035 | 0.429±0.026 |
> > > > | TimeMAE | 0.141±0.023 | 0.556±0.017 | 0.429±0.022 |
> > > > | UniTS | 0.308±0.057 | 0.610±0.105 | 0.475±0.119 |
> > > > | PatchTST (Mask) | **0.338±0.007** | 0.548±0.016 | 0.384±0.018 |
> > > > | PatchTST (CL) | 0.253±0.009 | 0.444±0.007 | 0.275±0.021 |
> > > > | PatchTST (PPT) | 0.263±0.023 | **0.644±0.048** | **0.481±0.050** |
> > > >
> > > > The result that PPT outperforms other SOTA methods in terms of NMI and ARI scores highlights its ability to capture the internal temporal structure and order relationships within the time series, leading to clustering results that align well with the true class labels. In summary, we show that PPT demonstrates its effectiveness in generating meaningful and informative representations for clustering tasks.

---

> > > > > ### Author Response · Authors · 2024-11-19
> > > > >
> > > > > **Q3. Limited evaluations. Experiments are all one-to-one dataset. The authors do not consider one-to-many, or other settings to test whether the representations are generalizable to cross-dataset settings.**
> > > > >
> > > > > We thank the reviewer for the valuable comments and the suggestion for further experiments. While we acknowledge the importance of assessing the transferability of representations in cross-datasets, we believe that it is not the primary focus of our current work.
> > > > >
> > > > > Our research is focused on introducing a novel time series specific pretext task, PPT, that leverages the order information of patches in time series data to enhance the performance of existing time series representation learning methods. The main objective we wanted to convey through out paper was the effectiveness of PPT in capturing order-aware and meaningful representations within a single dataset, specifically for tasks where order information is crucial.
> > > > >
> > > > > We also respectfully disagree that our paper has limited evaluations. We have conducted an extensive number of comparisons against several SOTA baselines in self-supervised linear probing, semi-supervised training, and supervised training, demonstrating the possibility and effectiveness of our newly proposed pretext task. We have also conducted extensive analysis on our patch shuffling strategies, and importance visualization by patches. We also conducted multiple experiments to validate our ACF-COS metric. We have also conducted additional experiments with UniTS, as the reviewer suggested.
> > > > >
> > > > > While cross-dataset evaluations can offer additional insights into the generalizability of the learned representations, we argue that they are not essential to validate the effectiveness of PPT in capturing order information. The primary goal of our work is to introduce a novel pretext task that can be integrated into existing patch-based time series representation learning methods to enhance their performance, rather than to develop a universal representation that can be transferred across datasets.
> > > > >
> > > > > Moreover, conducting cross-dataset evaluations would require careful consideration of the compatibility between the source and target datasets, with additional technical concerns such as time length, sampling frequency, etc., We believe that this is currently out of our paper’s scope, and we hope to explore this in our future research.
> > > > >
> > > > > In summary, while we appreciate the reviewer's suggestion to conduct cross-dataset evaluations, we believe that **our paper focuses on demonstrating the effectiveness of PPT within a single data, which is a crucial first step in validating the proposed pretext task.** We have conducted extensive one-to-one dataset experiments to evaluate the performance of PPT in comparison to other methods, conduct experiments on UniTS and clustering. We believe that these experiments provide a comprehensive assessment of PPT’s ability to capture meaningful representations and improve performance in time series classification tasks. Nevertheless, we acknowledge the importance of exploring the generalizability of the learned representations and plan to address this aspect in our future research.

---

> > > > > > ### Author Response · Authors · 2024-11-19
> > > > > >
> > > > > > **Q4. The gains are not convincing since many other baselines outperform the proposed method in several metrics.**
> > > > > >
> > > > > > Thank you for this insightful feedback. We acknowledge that some of the results of PPT have been outperformed by other baselines in metrics like Recall for Self-Supervised Linear probing and accuracy in Semi-Supervised Learning. However, we would like to cordially request the reviewer to take a closer look at the performance gain obtained by incorporating PPT into PatchTST and PITS.
> > > > > >
> > > > > > Both PatchTST and PITS have utilized mask-and-reconstruct and contrastive learning pretext tasks in their original implementations. By simply switching the pretext task to PPT, we observe a strong performance gain. For instance, in the EMO task, the F1 Score improves from 45% to 54% (+9%) for PatchTST, and Accuracy improves from 69% to 75% (+6%) for PITS. For the GL Task, the accuracy improves from 88% to 92% (+4%) for PatchTST and 87% to 92% (+5%) for PITS. In semi-supervised training, the gap is even larger, where we observe that PITS improves from 72% to 83% (+11%) in F1 Score in the 1% limited training data scenario.
> > > > > >
> > > > > > These improvements demonstrate the effectiveness of PPT as a pretext task in enhancing the performance of existing time series representation learning methods in tasks that exhibit order information such as human activity recognition, ECG, etc., (We also contribute ACF-COS, a novel metric that can pre-assess the utility of PPT).
> > > > > >
> > > > > > While we acknowledge that our work may not be the state-of-the-art (SOTA) in all metrics across all tasks, we believe that PPT provides a fresh perspective in the direction of representation learning for time series analysis. To the best of our knowledge, we are the first to self-supervise models based on the order information of patches in time series data. Given the increasing adoption of patch-based strategies in time series models, we believe that our contribution extends beyond metric performance and offers valuable insights to the time series community.
> > > > > >
> > > > > > In summary, while PPT may not outperform all baselines in every metric, the significant performance gains obtained by incorporating PPT into PatchTST and PITS demonstrate its effectiveness as a pretext task for time series representation learning.

---

> > > > > > > ### Author Response · Authors · 2024-11-19
> > > > > > >
> > > > > > > **Q5. It's not clear whether this method even works for diverse dataset pretraining settings as in UniTS (https://arxiv.org/abs/2403.00131). Since this method is mainly for pretext tasks, this is an important point to raise given we're in the era of foundation models.**
> > > > > > >
> > > > > > > Thank you for raising this important point. While we acknowledge the growing importance of foundation models and diverse dataset pretraining setups for time series, as demonstrated by works like UniTS, our work has a different focus and scope.
> > > > > > >
> > > > > > > The primary objective of our paper PPT is to introduce a novel pretext task for time series that leverages the order information of patches in time series data. Our pretext task can be incorporated into existing patch-based time series models like PatchTST and PITS, and have demonstrated their effectiveness in various experimental setups (supervised learning, self-supervised learning, etc.)
> > > > > > >
> > > > > > > Unlike UniTS, which aims to develop a unified framework for time series foundation models,**our work is not intended to be a comprehensive solution for diverse dataset pretraining**. Instead, our focus is on improving the performance of existing patch-based methods in time series applications where order information is important, by introducing a new pretext task that captures this order information.
> > > > > > >
> > > > > > > We believe that our contribution is valuable in advancing time series representation learning, particularly in tasks where order information is crucial. While foundation models and diverse dataset pretraining are undoubtedly important areas of research, our work targets a different aspect of time series analysis and contributes to this field in a complementary manner.

---

> > > > > > > > ### Author Response · Authors · 2024-11-27
> > > > > > > >
> > > > > > > > Dear Reviewer GFNC,
> > > > > > > >
> > > > > > > >
> > > > > > > > We appreciate the reviewer for the thorough reviews provided for our paper. As we approach the final days of the discussion period, we are writing regarding our work PPT. **We have devoted considerable time and effort to address the reviewer’s valuable feedback, conducting multiple additional experiments and providing comprehensive explanations** to ensure our novel contributions.
> > > > > > > >
> > > > > > > > In response to the reviewer’s suggestion, we have:
> > > > > > > > 1) Implemented comparisons with the latest state-of-the-art method (UniTS; NeurIPS 24)
> > > > > > > > 2) Performed clustering experiments to evaluate representation performance
> > > > > > > > 3) Addressed potential misunderstandings
> > > > > > > >
> > > > > > > >
> > > > > > > >
> > > > > > > >
> > > > > > > > Our work introduces a novel contribution to patch-based time series representation learning, where we believe we are the first to introduce order-aware self-supervision through our carefully designed patch-based pretext task. The results show that our approach is effective for tasks exhibiting both the temporal and channel order information. We have also provided additional metrics “ACF-CoS” to identify such datasets that exhibit order information, enabling us to pre-assess them prior to using our pretext task. We also note that PPT is compatible with many of the recent patch-based architectures in time series including PatchTST (Transformer), PITS (Linear), and even recent time series language models like GPT4TS.
> > > > > > > >
> > > > > > > >
> > > > > > > >
> > > > > > > >
> > > > > > > > We want to emphasize that our paper's primary **contribution lies in contributing to the time series community that order-based pretext tasks can be effectively applied to time series data, with promising gains**. Rather than positioning ourselves as achieving state-of-the-art performance, our goal is to open new research directions in this understudied area.
> > > > > > > >
> > > > > > > >
> > > > > > > >
> > > > > > > >
> > > > > > > > We thank the reviewer for providing helpful suggestions for improving our paper, and **we are eagerly awaiting the response to ensure that we can address any remaining concerns in our final rebuttal.**
> > > > > > > >
> > > > > > > >
> > > > > > > >
> > > > > > > >
> > > > > > > > Best regards,
> > > > > > > >
> > > > > > > >
> > > > > > > > Authors

---

> > > > > > > > > ### Comment · Reviewer_GFNC · 2024-11-27
> > > > > > > > > **Response to Rebuttals**
> > > > > > > > >
> > > > > > > > > I thank the authors for their rebuttals and discussion points. The rebuttals unfortunately have not convinced me about the limited evaluation setups (no one-to-many considered which makes it unfair since other prior works do have those), limited gains, lack of pretraining with multiple datasets, and comparisons with SoTA baselines. In fact, it has been shown in the UniTS paper that UniTS outperforms PatchTST, and other baselines, and while your work shows PatchTST can be enhanced in some settings with order aware pretext task, it's not clear what is best. Overall, in my opinion, this paper does not move the state of the art too far and is incremental in its current form. Therefore, I choose to respectfully keep my current score. Thanks.

---

### Official Review · Reviewer_Z8Ek · 2024-11-03

**Soundness:** 4
**Presentation:** 3
**Contribution:** 4
**Rating:** 8
**Confidence:** 4

**Summary:**

This paper introduces a new pretext task designed for patch-based time series models, focusing on the order of time series patches. The Patch order-aware Pretext Task (PPT) is aimed at improving time series classification performance by considering the sequential and channel-wise order of data patches. This task leverages controlled patch permutations to produce supervisory signals for training models to learn inherent order dependencies. PPT incorporates both patch order consistency learning and contrastive learning to handle weakly and strongly permuted patch sequences, achieving better model performance in various time series tasks.

**Strengths:**

- Novelty in time series pretext task: PPT introduces a novel pretext task that leverages patch order consistency learning and contrastive learning to improve time series classification performance. This is a significant contribution to the field of time series representation learning.
- Flexibility and adaptability: By introducing both consistency and contrastive learning, PPT can be incorporated into various task settings, either self-supervised or supervised, making it a versatile method for time series classification.
- New metric for evaluating the importance of orderness: The paper also introduces ACF-CoS, which measures the importance of orderness in time series data, providing a new metric for evaluating the effectiveness of PPT and potentially other time series models.
- Strong empirical results: The paper demonstrates the effectiveness of PPT in improving time series classification performance across various datasets, showing significant improvements over existing methods.
- Clear and well-structured presentation: The paper is well-written and structured, especially the explanation of the PPT task and its illustration with examples. The clarity of the presentation enhances the understanding of the proposed method.

**Weaknesses:**

- Unclear details on consistency loss: The paper do not provide more detail into how the context vectors are generated other than stating that they are generated by an auto-regressive model. More details into the architecture and training of this model would be helpful for better understanding the consistency loss.
- Lack of discussion on hyperparameter $\lambda$: The paper does not provide a detailed study or discussion on the hyperparameters $\lambda$, which are to be "learnable parameters". In the paper they cited that these hyperparameters are calculated from the prediction of a separate model, but more details on how this is done would be beneficial. It would also be helpful to provide ablation studies on the effect of such dynamic $\lambda$ on the performance of the model compared to fixed $\lambda$.

**Questions:**

1. Could you provide more details on how the context vectors are generated by the auto-regressive model for the consistency loss? How is the architecture of this model designed, and how is it trained?
2. Could you provide more details on how the hyperparameters $\lambda$ are adjusted during training? What are the final results of the learned $\lambda$ values at the end of training, and how do they affect the performance of the model?

---

> ### Author Response · Authors · 2024-11-19
>
> We sincerely appreciate the positive review and the detailed description provided by the reviewer to improve our manuscript. We are encouraged that the reviewer has recognized our work's novelty, flexibility, and strong empirical performance. We particularly appreciate the reviewer's acknowledgment of ACF-COS, our proposed metric for quantifying order information in time series, and agree with the assessment of its potential benefits for broader time series applications.
>
> In response to the reviewer's feedback, we have made the following improvements:
>
> 1. Updated our main manuscript (Line 255-258) to further incorporate additional details on how the context vectors are generated, and have added Section **F.3: Autoregressive models** in the Appendix for further explanations.
>
> 2. Performed and reported (**Appendix G.4 Learnable and Fixed Lambda**) analysis on the learned hyperparameters $\lambda$ and compared it against fixed hyperparameter setups (25 combinations). We also updated our main manuscript (**Lines 515-516**) to incorporate this information. Briefly, the learnable hyperparameter approach achieves comparable performance to the optimal fixed hyperparameter setup, effectively eliminating the need for extensive parameter searches.
>
> Here, we provide detailed answers to each of the questions.

---

> > ### Author Response · Authors · 2024-11-19
> >
> > **Q1. Could you provide more details on how the context vectors are generated by the auto-regressive model for the consistency loss? How is the architecture of this model designed, and how is it trained?**
> >
> > A1. Thank you for your question, which helped us realize that the current manuscript lacked details on the auto-regressive model used for generating context vectors in the consistency loss. We appreciate the opportunity to provide more information about this important component of our approach.
> >
> > The consistency loss is employed on the sequence of patches in both time and channel order. To generate the context vectors, we use a single-layer unidirectional LSTM that takes the patch sequences as input. The final cell state of the LSTM serves as the context vector, effectively summarizing the sequence of patches.
> >
> > For consistency learning, these context vectors are assigned pseudo labels: 1 if the sequence of patches is from the original sequence, and 0 if the sequence of patches is from a strongly shuffled sequence. Initially, the auto-regressive model is unable to recognize the correct sequence. However, through end-to-end training alongside the backbone patch model, the model learns to discriminate between the original and strongly shuffled patch sequences.
> >
> > The consistency learning is applied along both the time and channel dimensions. For the time dimension, the module learns to recognize the correct order of patches that should occur temporally. For the channel dimension, the module learns the cross-channel relationships of patches. As shown in **Fig1:Motivation**, we can observe distinct patterns co-occuring along the channel dimension, and learning this cross-channel relationship has shown to enhance model performance.
> > Based on the review, we were able to update our main manuscript by adding the following explanations (**Line 255-258**)
> >
> > ```markdown
> > Here, we employ single-layer uni-directional LSTMs that process the patch sequences. The final cell state of each LSTM serves as the context vector, effectively summarizing the patch sequences. These LSTM modules are trained end-to-end with the backbone model (details in Appendix F)
> > ```

---

> > > ### Author Response · Authors · 2024-11-19
> > >
> > > **Q2. Provide more details on how the Hyperparameters $\lambda$ are adjusted during training.**
> > >
> > > A2. We are happy to have the opportunity to further explain how the hyperparameters $\lambda$ are adjusted during training. This strategy was first proposed by Kendall et al. [1] and Liebel and Körner [2] and has been consequently adopted in the time series community [3, 4].
> > >
> > > It has been shown that simply aggregating losses from multiple tasks can lead to negative transfer, as training multi-task models requires a balance between task-specific losses. A naive solution would be to perform a grid search over all possible combinations of weights, but the search space grows exponentially, and the weights are fixed throughout the whole training process, not being able to adapt to the training dynamics [4].
> > >
> > > As such, the learnable loss function [1, 2] is based on the uncertainty-based method that maximizes the Gaussian likelihood by considering each task's homoscedastic uncertainty. The learnable parameters are jointly optimized using the task-specific losses, with the parameters being regularized so as not to become too small during training.
> > >
> > > In our experiment, we adopt this learnable loss strategy to automatically balance the consistency loss and margin loss. Specifically, we introduced learnable parameters $\lambda_1$ and $\lambda_2$ for the consistency loss and margin loss, respectively. These parameters are initialized with a value of 1.0 and are optimized alongside the model parameters during training.
> > >
> > > During training, the model parameters and the learnable loss parameters are updated simultaneously using backpropagation. This allows the model to adaptively adjust the importance of each loss term based on the training dynamics and the uncertainty of each task. Based on the reviewer's suggestion, we have visualized the training dynamics of the learnable lambdas in **Appendix G.4**. We observe that while the parameters were initialized with a value of 1.0, they adjusted their values throughout the training and converged to 0.39 (Consistency) and 0.60 (Margin) for PITS and 0.18 (Consistency) and 0.23 (Margin) for PatchTST, showcasing that the magnitude of the learned hyperparameters may differ for each model.
> > >
> > > [1] Kendall, Alex, Yarin Gal, and Roberto Cipolla. "Multi-task learning using uncertainty to weigh losses for scene geometry and semantics." Proceedings of the IEEE conference on computer vision and pattern recognition. 2018.
> > >
> > > [2] Liebel, Lukas, and Marco Körner. "Auxiliary tasks in multi-task learning." arXiv preprint arXiv:1805.06334 (2018).
> > >
> > > [3] Dong, Jiaxiang, et al. "Simmtm: A simple pre-training framework for masked time-series modeling." Advances in Neural Information Processing Systems 36 (2024).
> > >
> > > [4] Kim, Jaeho, et al. "Multitask Deep Learning for Human Activity, Speed, and Body Weight Estimation Using Commercial Smart Insoles." IEEE Internet of Things Journal 10.18 (2023): 16121-16133.

---

> > > > ### Author Response · Authors · 2024-11-19
> > > >
> > > > **Q3. What are the final results of $\lambda$ and how do they affect model performance?**
> > > >
> > > >
> > > > A3. We thank the reviewer for this insightful question. To answer this, we conducted multiple fixed-$\lambda$ experiments and compared them with the results from the learnable $\lambda$ setup. Briefly, we show that the learned $\lambda$s are the best performing compared to the fixed $\lambda$ setup.
> > > >
> > > > We conducted a grid search between $\lambda_{consistency} = [0.1, 0.2, 0.5, 1.0, 2.0]$ and $\lambda_{margin} = [0.1, 0.2, 0.5, 1.0, 2.0]$, resulting in 25 different combinations. We performed our experiments on the PITS (Linear) model and PatchTST (Transformer) with the PTB dataset. The following table shows the F1 Score (average of 5 random seeds) for the top 3 and bottom 1 performing combinations for PITS:
> > > >
> > > > | Rank | Model | Consistency | Margin | F1 |
> > > > |------|-------|-------------|--------|------|
> > > > | 1    | PITS  | Learnable   | Learnable | 91.24 ± 0.72 |
> > > > | 2    | PITS  | 0.1         | 1.0    | 91.20 ± 0.66 |
> > > > | 3    | PITS  | 0.1         | 0.1    | 91.00 ± 0.79 |
> > > > | ...  | ...   | ...         | ...    | ... |
> > > > | 25   | PITS  | 0.1         | 0.5    | 89.23 ± 1.38 |
> > > >
> > > > Similarly, for PatchTST:
> > > >
> > > > | Rank | Model | Consistency | Margin | F1 |
> > > > |------|-------|-------------|--------|------|
> > > > | 1    | PatchTST | Learnable   | Learnable | 88.67 ± 0.95 |
> > > > | 2    | PatchTST | 1.0         | 0.1    | 88.57 ± 0.78 |
> > > > | 3    | PatchTST | 0.5         | 0.1    | 88.39 ± 0.81 |
> > > > | ...  | ...   | ...         | ...    | ... |
> > > > | 25   | PatchTST | 0.1         | 0.1    | 85.61 ± 1.22 |
> > > >
> > > > We show that the learnable hyperparameters are comparable to the best-performing fixed-hyperparameters that are manually searched, showcasing that the learnable hyperparameter setup can significantly reduce hyperparameter tuning while leading to optimal performance.
> > > >
> > > > We report the full results of the fixed grid search, and also visualize how the hyperparameters are adjusted during the model training for our setup in **Appendix G.4.**  We have also updated the main manuscript (**Lines 515-516**) to incorporate this information as follows:
> > > >
> > > > ```markdown
> > > > **Learnable and Fixed Loss Coefficients** We conducted experiments with fixed hyperparameters $\lambda_1$ and $\lambda_2$ to compare against the learnable strategy adopted in our work. The results and analysis of these experiments can be found in Appendix G.
> > > > ```
> > > >
> > > > We thank the reviewer for the detailed discussion of our works and suggesting experiments that have highly helped us improve our work. If there are additional explanations that are needed, please do not hesitate to let us know, so that we can further improve our scores. Thank you for your services.

---

> > > > > ### Author Response · Authors · 2024-11-27
> > > > >
> > > > > Dear Reviewer Z8EK,
> > > > >
> > > > >
> > > > > We thank the reviewer for providing constructive feedback and highlighting our work as **"a significant contribution to the field of time series representation learning"**. We are delighted and motivated by your appraisals.
> > > > >
> > > > >
> > > > > As we now approach the end of the discussion period, we kindly ask if there are any uncertainties left that we could address, as we would be more than happy to provide additional clarifications and explanations if needed.
> > > > >
> > > > >
> > > > > We thank you once again.
> > > > >
> > > > >
> > > > > Best regards,
> > > > >
> > > > > Authors

---

> > > > > > ### Comment · Reviewer_Z8Ek · 2024-11-27
> > > > > >
> > > > > > Dear authors,
> > > > > >
> > > > > > Thank you for your revision. These address my concerns. However, I think the current rating is appropriate, so it will remain unchanged.

---

> > > > > > > ### Author Response · Authors · 2024-11-28
> > > > > > >
> > > > > > > Dear **Reviewer Z8Ek**
> > > > > > >
> > > > > > > Thank you for the constructive review and for helping us improve our work! We are happy to see that we have addressed the reviewer's concerns and have kept the rating towards acceptance.
> > > > > > >
> > > > > > > Thank you!
> > > > > > >
> > > > > > > Authors.

---

### Official Review · Reviewer_o6fk · 2024-11-03

**Soundness:** 2
**Presentation:** 3
**Contribution:** 2
**Rating:** 6
**Confidence:** 4

**Summary:**

The paper introduces a novel Patch Order-aware Pretext Task (PPT) for time series classification, focusing on the importance of patch order in time series data. Traditional patch-based models often overlook the critical order dependencies in time and channel dimensions, which can lead to suboptimal performance. PPT addresses this gap by generating supervisory signals through controlled channel-wise patch permutations, allowing models to learn time series' intrinsic sequential characteristics. The main contributions of the paper are as follows: (1) Patch Order-aware Pretext Task (PPT): This is the first pretext task specifically designed to enhance order-awareness in patch-based time series models. PPT leverages controlled channel-wise permutations to provide structural supervisory signals in both time and channel dimensions. (2) Two Learning Methods: Patch Order Consistency Learning evaluates the correctness of patch order by distinguishing between original and strongly permuted sequences; Contrastive Learning differentiates between weakly and strongly permuted sequences, helping the model capture finer order distinctions. (3) ACF-COS Metric: The paper proposes ACF-COS (Autocorrelation Function with Cosine Similarity), a metric to quantify the "orderness" in time series data, which serves as a pre-evaluation tool to determine if PPT would be beneficial for a particular dataset. (4) Performance Gains: PPT demonstrates significant performance improvements on various time series tasks, especially in scenarios where order-awareness is crucial, such as ECG and human activity recognition (HAR) tasks, with performance boosts of up to 7% and 5% respectively. By integrating order-awareness through PPT, the approach enhances the ability of time series models to utilize structural dependencies across patches, setting a new direction for patch-based time series analysis.

**Strengths:**

**Introduction of the Patch Order-aware Pretext Task (PPT)**: This paper presents the first order-aware pretext task specifically designed for patch-based time series models. Unlike traditional tasks such as masking and reconstruction, PPT captures essential order dependencies across time and channel dimensions, a distinctive feature of time series data. By leveraging this natural inductive bias of sequence ordering, PPT provides effective self-supervision signals that help the model better understand structural characteristics in time series classification tasks.

**Implementation of Order Consistency and Contrastive Learning for Pretraining**: To achieve order-aware learning within PPT, the authors designed two learning strategies: order consistency learning and contrastive learning. Order consistency learning distinguishes between the original sequence and strongly perturbed sequences, validating the model's understanding of patch order correctness. Contrastive learning differentiates between weakly and strongly perturbed sequences, allowing the model to capture finer distinctions in order dependencies. Together, these learning approaches enhance the model's sensitivity to sequence order, helping it capture the critical ordering information embedded within time series data and improving performance on order-dependent tasks.

**Significant Performance Improvement**: PPT demonstrates notable improvements across multiple time series tasks, particularly in scenarios where order information is critical. Experimental results show that PPT enhances accuracy by up to 7% in ECG classification and by 5% in human activity recognition (HAR), underscoring its effectiveness. By integrating PPT in both self-supervised and supervised settings, models are able to better leverage the structural dependencies in time series data, achieving superior classification performance. This advancement not only showcases PPT's value in practical applications but also highlights the potential of order-aware tasks in time series analysis.

**Weaknesses:**

**Limited Applicability to Non-Order-Dependent Tasks**: PPT is specifically designed to exploit order dependencies in time series data, which limits its applicability to tasks where sequence order is critical. For datasets or tasks that do not rely heavily on the sequential order of data (e.g., tasks with significant noise or random sequences), PPT may add unnecessary complexity without meaningful performance gains. The paper itself notes that PPT does not perform well in such scenarios, and a robust evaluation metric is required to assess its suitability beforehand.

**Dependence on Hyperparameter Tuning for ACF-COS and Permutation Strength**: The effectiveness of PPT depends on the selection of appropriate hyperparameters, especially for the ACF-COS metric and permutation intensity (weak vs. strong permutations). However, the paper provides limited guidance on systematically tuning these parameters for different datasets, which could hinder reproducibility and practical implementation. Users may need to perform extensive trial-and-error testing to optimize PPT for specific datasets, which is resource-intensive and may not always yield consistent results.

**Limited Success in Time Series Forecasting**: Although PPT shows promise in classification tasks, it is less effective for time series forecasting applications, where trend continuity is crucial. The patch permutation strategy disrupts these trends, making it challenging for models to capture long-term dependencies needed for accurate forecasting. The paper does not provide a modified approach for forecasting tasks, which restricts PPT’s utility and impact in a broader range of time series applications.

**Questions:**

1. **Is there a method that can be adapted for both classification and forecasting tasks in time series?** Specifically, is it possible to design a pretext task that effectively utilizes order information for classification while preserving trend continuity to support forecasting tasks? If so, can this approach balance prediction accuracy with the advantages of sequence dependency?

2. **Does the ACF-COS metric demonstrate robust and broad applicability in measuring the orderness of time series data?** Specifically, can ACF-COS accurately differentiate between various types of time series tasks, such as high-noise sequences, non-order-dependent tasks, and strongly order-dependent tasks? Additionally, how can we ensure the stability of ACF-COS across different patch sizes, sampling frequencies, or data distributions to reliably evaluate the suitability of PPT for diverse time series datasets?

---

> ### Author Response · Authors · 2024-11-19
>
> We thank the reviewer for the thoughtful review of our work. We appreciate that the reviewer has thoroughly gone through our paper, highlighting the novelty of the two learning methods proposed in this work, pinpointing the performance gain obtained through the use of PPT, and the utility of ACF-COS metrics. Here, we have prepared detailed responses to the reviewer’s feedback and questions. Please let us know if there are additional explanations that we can provide to update our scores!

---

> > ### Author Response · Authors · 2024-11-19
> >
> > **Q1. Is there a method that can be adapted for both classification and forecasting tasks in time series? Specifically, is it possible to design a pretext task that effectively utilizes order information for classification while preserving trend continuity to support forecasting tasks? If so, can this approach balance prediction accuracy with the advantages of sequence dependency?**
> >
> > Thank you for this insightful question! While our current manuscript focuses explicitly on time series classification tasks (please kindly refer to our Keywords in the Submission details), our future research direction is to design pretext tasks that can be adapted for both the classification and forecasting setups. We found that PPT is highly effective for classification tasks where order information is crucial, but it results in subpar model performance for forecasting tasks, as noted in our limitations. This is because shuffling the patch order disrupts the "trend" factor, which is an essential criterion for forecasting.
> >
> > As the reviewer has suggested, a possible direction is to jointly utilize the mask-based pretext task and our order-aware pretext task, ensuring that the trend continuity is preserved for forecasting tasks. We believe that this approach may strike a balance between improving the prediction of forecasting while supervising order information in patches.
> >
> > In our future work, we plan to explore this direction by developing a hybrid pretext task that combines the strengths of both mask-based and order-aware approaches. By doing so, we aim to create a more versatile pretext task that can be effectively applied to both classification and forecasting tasks, thereby expanding the scope and impact of our research.
> >
> > We appreciate the reviewer's valuable suggestion and look forward to investigating this promising direction in our future research efforts.

---

> > > ### Author Response · Authors · 2024-11-19
> > >
> > > **Q2. Does the ACF-COS metric demonstrate robust and broad applicability in measuring the orderness of time series data Specifically, can ACF-COS accurately differentiate between various types of time series tasks, such as high-noise sequences, non-order-dependent tasks, and strongly order-dependent tasks? Additionally, how can we ensure the stability of ACF-COS across different patch sizes, sampling frequencies, or data distributions to reliably evaluate the suitability of PPT for diverse time series datasets?**
> > >
> > > Thank you for providing us the opportunity to further explain our proposed metric ACF-COS. As the reviewer mentioned, PPT was originally designed for time series classification tasks that exhibit order information. PPT exploits the time and channel-wise patch order dependency and supervises this information for model training. However, not all time series exhibit order information, and some tasks may have class labels independent of these temporal or channel-order dependencies. In such tasks, PPT may not be suitable as a pretext task.
> > >
> > > To pre-assess such tasks prior to deploying PPT as a pretext task, we proposed ACF-COS, where a high ACF-COS score indicates that order is more important, and a low ACF-COS score indicates that order is less important in the task. ACF-COS is calculated by constructing the autocorrelation vectors of the original and patch-permuted sets of the temporal sequence. A high cosine similarity indicates that permuting these patches did not disrupt the order information, while a low cosine similarity indicates that order information was preserved despite permuting the patch sequences.
> > >
> > > From our extensive experiments conducted on 24 different classification datasets, we showed that ACF-COS is robust in measuring the orderness of time series. These datasets consist of a wide range of tasks where order is not important (White Noise; ACF-COS=0.001), less important (GestureMidAirD1; ACF-COS=0.186), mid important (Cricket; ACF-COS=0.418) and heavily important (Step Function; ACF-COS=0.902). The ACF-COS well identifies the order characteristics in diverse time series tasks (both real-world and synthetic). We also fitted two least square methods indicating the correlation between the ACF-COS score and the performance gain obtained from PPT is positive with a significant p-value.
> > >
> > > To mitigate the effect of patch sizes in permuting the sequence, we have reported the results on 3 different patch sizes for each time series dataset and averaged their results for a robust metric. As such, ACF-COS is stable and reliable in evaluating the suitability of PPT for time series tasks.
> > >
> > > In summary, our work proposes ways to supervise order information for time series classification tasks while also providing a pre-assessment tool for the effectiveness of PPT, making it a comprehensive contribution to the field. We acknowledge that further research is needed to explore the stability of ACF-COS across different sampling frequencies and data distributions, and we plan to address these aspects in our future work.

---

> > > > ### Author Response · Authors · 2024-11-27
> > > >
> > > > Dear Reviewer o6fk,
> > > >
> > > > We thank the reviewer for the positive evaluation of our work. We have provided detailed explanations addressing the reviewer’s questions. As we now approach the end of the discussion period, we cordially request that the reviewer review our responses. We hope we have adequately addressed the reviewer’s concerns and would appreciate the consideration of revising our scores if our answers are satisfactory. **Please don’t hesitate to ask if any uncertainties remain, as we are committed to improving our paper further.**
> > > >
> > > > Best regards,
> > > >
> > > > Authors

---

> > > > > ### Author Response · Authors · 2024-12-03
> > > > >
> > > > > Dear **Reviewer o6fk,**
> > > > >
> > > > > As the dues are coming, we kindly request the reviewer to have a look at our responses. We would appreciate it if the reviewer could revise the scores based on our answers, and ask any questions that could help clarify any uncertainties left behind.
> > > > >
> > > > > Thank you.
> > > > >
> > > > > Authors.

---

### Official Review · Reviewer_i3b6 · 2024-11-03

**Soundness:** 3
**Presentation:** 4
**Contribution:** 3
**Rating:** 6
**Confidence:** 4

**Summary:**

The authors believe that patch order awareness is very important in time series modeling and  introduce PPT (Patch order-aware Pretext Task) for time series classification. The authors also develop ACF-CoS metric for quantifying the importance of patch orderness and interpreting applicable to PPT scenarios. Experimental results demonstrate that PPT shows advantages in self-supervised, fully-supervised, and semi-supervised learning with sparse labels.

**Strengths:**

- Work is novel and motivation regarding patch order is reasonable for time series analysis.
- Well written, clear structure, easy to understand
- Experiments are sufficient, and the analysis based on ACF-CoS is interesting

**Weaknesses:**

Rocket[1] and MiniRocket[2] are important baselines for time series classification tasks. I think considering them can make the experiment more comprehensive (including classification performance and algorithm efficiency).

[1] Angus Dempster, et al. ROCKET: Exceptionally fast and accurate time series classification using random convolutional kernels. arxiv preprint arxiv: 1910:13051

[2] Angus Dempster, et al. MINIROCKET: A Very Fast (Almost) Deterministic Transform for Time Series Classification. arxiv preprint arxiv: 2012:08791

**Questions:**

- Time dimension patch order is intuitive, but channel dimension order is not easy to understand and it needs further discussion.
- Although the author provides the consumed time of permuting a batch of instances, considering the need to store additional augmented samples and the need for additional calculation, it would be better if the author can provide the additional speed and memory consumption required by PPT.
- Confusing content: In Tab. 4 caption: “a decrease in ACF-CoS leads to an improved likelihood of PPT being beneficial”. Should “a decrease” be modified to “an increase”? In my understanding, there is a positive correlation between the ACF-COS score and PPT's performance.

---

> ### Author Response · Authors · 2024-11-19
> **Overall Changes Summary.**
>
> Thank you for your thorough and constructive feedback on our work! We are grateful that the reviewer has recognized both the novelty and clarity of our paper, as well as acknowledged the soundness of our motivation for using patch order supervision as a pretext task in time series classification and finding our experiments sufficient.
> Based on the reviewer’s suggestion, we have
> 1. Conducted additional baseline comparisons,
> 2. Updated the main manuscript (**lines 189-192**) to incorporate additional details of channel order learning to make our explanation more intuitive.
> 3. Reported the speed and memory consumption needed in our work in **Section F.2: Time and Memory Cost of PPT** in the Appendix
> 4. Modified the typo in our table caption.
>
> Below, we have provided detailed responses. Please don't hesitate to seek any additional clarification needed.

---

> > ### Author Response · Authors · 2024-11-19
> >
> > **W1. Consider Rocket and MiniRocket as baselines in our work.**
> >
> > A. We thank the reviewer for providing insightful suggestions and highly agree with the reviewer’s opinion that incorporating additional baselines could make our experiment more comprehensive. As such, we have performed MiniRocket (successor of Rocket) and UniTS (the most recent SOTA work accepted to NeurIPS 2024 as suggested by reviewer GFNC) experiments with self-supervised linear probing on the PTB tasks, where we report the results below. MiniRocket shows competitive performance compared to SOTA methods, but is still outperformed in key metrics.
> >
> > | Model | Accuracy | F1 score | AUROC | AUPRC |
> > |-------|----------|-----------|--------|--------|
> > MiniRocket | 85.52±2.17 | 90.19±1.67 | 88.55±2.42 | 93.74±2.94 |
> > | UniTS* | 84.20±1.01 | 89.57±0.60 | 88.98±1.35 | 94.81±0.84 |
> > | PITS (+PPT) | **86.48±0.40** | **91.24±0.26** | **91.83±1.36** | **96.26±0.85** |

---

> > > ### Author Response · Authors · 2024-11-19
> > >
> > > **Q1. Time Dimension patch order is intuitive, but channel order is not easy to understand.**
> > >
> > > A1. We thank the reviewer for bringing this to our attention. While the time dimension patch order focuses on the temporal order dependency within each channel, the channel order captures cross-channel patterns that occur simultaneously at each time point (patch index). The channel order captures how different sensors (channels) appear together across channel dimensions at the same time point, allowing the model to learn cross-channel relationships. For instance, as shown in **Figure 1: Motivation**, we observe peaks occurring in all sensors (accelerometer and force sensors) at the same time point. However, when we permute the top two patches of the accelerometer sensor, these peaks do not co-align at the same time points as they did before, effectively disrupting the original channel order. As such, PPT utilizes this disruption in order information to self-supervise the model.
> > >
> > > To further enhance our explanation, we have updated the main manuscript by adding the following explanation in lines 189-192:
> > >
> > > ```markdown
> > > While time order $\boldP_{(:,d)}$ captures the temporal evolution within individual channels, the channel order sequence $\boldP_{(i,:)}$ represents a snapshot of patterns that appear together across all channels at a single time point, enabling the model to learn both temporal dynamics and concurrent cross-channel relationships.
> > > ```
> > > Please kindly let us know if the explanation can be further improved! Thank you once again.

---

> > > > ### Author Response · Authors · 2024-11-19
> > > >
> > > > **Q2. Considering the need to store additional augmented samples and the need for additional calculation.**
> > > >
> > > > A2. We thank the reviewer for raising this point. Here, we respectfully highlight a possible misunderstanding regarding the need to store additional augmented samples for PPT training. PPT **does not require explicit storage** of the augmented samples; instead, it only stores the shuffled patch indexes. During each run, these indexes are called and used to re-order the patches accordingly (In practice, we utilize *torch.gather()* to re-order the patches).
> > > > This approach offers two key advantages. First, we can reuse the same shuffled indexes for various samples, enabling us to create diverse augmented shuffled sets. Second, it drastically reduces the storage costs in directly storing raw augmented samples and minimizes the overhead of shuffling patches on the fly for every batch of samples, making PPT an effective technique.

---

> > > > > ### Author Response · Authors · 2024-11-19
> > > > >
> > > > > **Q3. Can the authors provide the additional speed and memory consumption required by PPT.**
> > > > >
> > > > > A3. We appreciate the reviewer's question regarding the additional speed and memory consumption required by PPT. We have conducted further experiments on the PTB task (Channel size of 15, Batch Size of 32, and 30 Patches for each channel) for self-supervised linear probing and have compared our method against mask-based and contrastive-based approaches. The detailed results of training time and memory consumption are reported in **Appendix F.2.**
> > > > >
> > > > > In summary, PPT employs additional auto-regressive models for constructing context vectors in the training process, slightly increasing the speed and memory consumption compared to previous pretext task methods. However, in the inference stage, these auto-regressive models can be discarded, and only the backbone model is used. In terms of time complexity, mask-based approaches took 70.3 seconds for the whole training, contrastive-based approaches took 92.9 seconds, and PPT took 139.5 seconds. This increased training time is due to the additional computations required for auto-regressive models constructing context vectors. Regarding memory complexity, the mask-based approach had a total of 79.7K parameters, the contrastive-based approach had 79.3K parameters, and PPT had 212K parameters. The higher number of parameters in PPT is primarily attributed to the use of auto-regressive models.
> > > > >
> > > > > Despite the increased computational overhead, we believe that the improved performance achieved by PPT, as demonstrated in our experimental results, justifies the use of this technique. We acknowledge that the current implementation of PPT has room for optimization, and we propose that exploring more efficient model designs for constructing context vectors (e.g., reducing representation dimensions, and re-using the auto-regressive models) would mitigate the computation burden.

---

> > > > > > ### Author Response · Authors · 2024-11-19
> > > > > >
> > > > > > **Q4. "Decrease" in Tab. 4 caption should be "Increase"**
> > > > > >
> > > > > > A4. We sincerely appreciate the reviewer’s detailed review of our manuscript. This was indeed a typo that we have now corrected:
> > > > > >
> > > > > > Original: "Here, a decrease in ACF-COS leads to an improved likelihood of PPT being beneficial."
> > > > > >
> > > > > > Revised: "Here, an **increase** in ACF-COS leads to an improved likelihood of PPT being beneficial."
> > > > > >
> > > > > > Thank you again for your valuable feedback that has helped improve the quality of our paper! We truly appreciate the time and effort spent by the reviewer. If there are additional details or explanations that we can provide, please kindly let us know what we can do more to update our scores.

---

> > > > > > > ### Comment · Reviewer_i3b6 · 2024-11-19
> > > > > > >
> > > > > > > I really appreciate your efforts during the rebuttal period and most of my concerns have been addressed. In the comparison with MiniRocket, although PPT slightly outperforms in terms of key metrics (classification performance), I would prefer to see a comparison with MiniRocket in terms of running time (efficiency).

---

> > > > > > > > ### Author Response · Authors · 2024-11-19
> > > > > > > >
> > > > > > > > **I really appreciate your efforts during the rebuttal period and most of my concerns have been addressed. In the comparison with MiniRocket, although PPT slightly outperforms in terms of key metrics (classification performance), I would prefer to see a comparison with MiniRocket in terms of running time (efficiency).**
> > > > > > > >
> > > > > > > > We greatly appreciate the reviewer’s recognition of our efforts during the rebuttal period and are pleased to know that most of the concerns have been addressed. Regarding the comparison between MiniRocket and PPT in terms of the “running time efficiency”, we acknowledge that MiniRocket is indeed a highly efficient algorithm due to its use of deterministic convolution kernels [1].
> > > > > > > >
> > > > > > > > However, It is important to note that PPT is designed to be incorporated into time series patch-based self-supervised representation learning, where much of the current time series research has been focusing on [2,3,4]. We point out that PPT can be adapted to many of the patch-based architectures, such as PatchTST and PITS, which is trained end-to-end, with mini-batch updates, while MiniRocket is unable to provide such functionality. This fundamental difference in architecture and training process explains the longer training time of PPT compared to MiniRocket.
> > > > > > > >
> > > > > > > > To provide a quantitative comparison, we measured the training times for both MiniRocket and PatchTST (+PPT) on the PTB task. MiniRocket took 29.83±9.16 seconds using an Intel(R) Xeon(R) Gold 6338 CPU @ 2.00GHz, while PatchTST (+PPT) required 139.5±10.76 seconds using a single NVIDIA RTX A6000 GPU.
> > > > > > > >
> > > > > > > > While we acknowledge the longer training time of PPT, we would like to emphasize that the primary goal of PPT is to introduce a novel pretext task that enhances the performance of patch-based deep architectures using order information between patches, and the trade-off between efficiency and performance is a compromise in this context.
> > > > > > > >
> > > > > > > > We acknowledge that there is room for further optimization of PPT, such as introducing more efficient auto-regressive models for patch sequence understanding or thinking of ways we can construct context vectors without these auto-regressive models, which could potentially reduce the training time.
> > > > > > > >
> > > > > > > > In summary, while MiniRocket is indeed a highly efficient time series algorithm, PPT is designed for general, patch-based deep architectures, which achieves improved classification performance compared to existing time series pretext tasks, which is the main focus of our work. We thank the reviewer's feedback and the opportunity to clarify the efficiency aspect of our work.
> > > > > > > >
> > > > > > > > [1] Angus Dempster, et al. ROCKET: Exceptionally fast and accurate time series classification using random convolutional kernels. arxiv preprint arxiv: 1910:13051
> > > > > > > >
> > > > > > > > [2] Lee, Seunghan, Taeyoung Park, and Kibok Lee. "Learning to Embed Time Series Patches Independently." The Twelfth International Conference on Learning Representations.
> > > > > > > >
> > > > > > > > [3] Cao, Defu, et al. "TEMPO: Prompt-based Generative Pre-trained Transformer for Time Series Forecasting." The Twelfth International Conference on Learning Representations
> > > > > > > >
> > > > > > > > [4] Jin, Ming, et al. "Time-LLM: Time Series Forecasting by Reprogramming Large Language Models." The Twelfth International Conference on Learning Representations.

---

> ### Comment · Reviewer_i3b6 · 2024-11-20
>
> Thank you for your responses. From the experiment results, I think MiniRocket is an extremely lightweight (can run on CPU), extremely efficient (nearly 4+ times faster than PatchTST+PPT), and very competitive time series classification algorithm. In my opinion, currently it is difficult to find a method that can compete with MiniRocket in terms of both performance and efficiency for time series classification task, which limits the application value of many so-called SOTA deep learning methods. However, given that PPT reveals time series characteristics from a novel perspective to enhance the performance of deep models, I tend to keep my score. I suggest that PPT can be expanded to more time series tasks, learn more general time series representations, and improve the actual application value in the real world.

---

> > ### Author Response · Authors · 2024-11-20
> > **Thank you for the thorough review**
> >
> > Thank you for the thorough review and insightful discussions throughout this review process. We greatly **appreciate the reviewer’s decision to maintain the rating towards acceptance, recognizing the novel perspective that PPT brings to time series analysis**. As the reviewer has suggested, we will look into expanding PPT in other applications in our future work. We are grateful for the enlightening discussions we have had and the valuable feedback you have provided.

---

### Author Response · Authors · 2024-11-25
**General Response**

We sincerely thank all reviewers for their thorough evaluation of our work.

We appreciate the positive feedback regarding our work PPT, which was described as **“novel, well-written, and easy to understand, a significant contribution to the field of time series representation learning” with “sufficient” experimentation and “strong empirical results.”** PPT is indeed a novel self-supervised learning algorithm designed to enhance existing patch-based methodology in time series.  We are also encouraged that reviewers have recognized the potential impact of our ACF-COS metric in benefiting other time series models.

Based on the reviews provided, we have **conducted 3 new experiments and included an additional baseline**, where we have accordingly updated our manuscript. In detail, **we performed linear probing experiments with UniTS, hyperparameter ablations with a fixed coefficient setup to compare it against our learnable coefficient setup, and a clustering comparison with several SOTA works**. We have updated our manuscripts to incorporate additional details as suggested by the reviewers.

Thank you once again for the constructive reviews and commentary. We believe our revisions comprehensively address the raised concerns. We remain available to provide any additional clarification. As some reviewers have not yet responded, we would greatly value their feedback to further improve our work.

Thank you very much!

---

### Comment · Area_Chair_sTyh · 2024-11-26
**Encouragement to Actively Participate in the Discussion Phase**

Dear Reviewers,

Thank you for your valuable contributions to the review process so far. As we enter the discussion phase, I encourage you to actively engage with the authors and your fellow reviewers. This is a critical opportunity to clarify any open questions, address potential misunderstandings, and ensure that all perspectives are thoroughly considered.

Your thoughtful input during this stage is greatly appreciated and is essential for maintaining the rigor and fairness of the review process.

Thank you for your efforts and dedication.

---

### Meta-Review · Area_Chair_sTyh · 2024-12-20

**Metareview:**

(a) Summary of Scientific Claims and Findings
The paper introduces Patch Order-aware Pretext Task (PPT), a novel self-supervised learning methodology tailored for time series classification. The key contributions include:
Patch Order Awareness: PPT supervises models using patch permutations, exploiting both time and channel-wise order dependencies in time series data.
Two Learning Methods: Patch Order Consistency Learning evaluates patch order correctness, while Contrastive Learning differentiates between weakly and strongly permuted sequences.
ACF-CoS Metric: A novel metric to quantify the importance of order dependencies in datasets, guiding the applicability of PPT.
Performance Gains: PPT improves classification accuracy by up to 7% on ECG tasks and 5% on human activity recognition tasks, outperforming masking-based pretext tasks.

(b) Strengths of the Paper
Novel and Well-Motivated Methodology: The focus on patch order awareness provides a unique perspective for time series representation learning, addressing a gap in current self-supervised methods.
Strong Empirical Results: Demonstrates consistent performance improvements across various tasks and datasets, with thorough experiments validating the efficacy of PPT.
ACF-CoS Metric: The introduction of a dataset-specific metric adds significant value by enabling pre-evaluation of PPT’s effectiveness.
Clear Writing and Presentation: The paper is well-structured and communicates complex ideas effectively, supported by comprehensive illustrations and examples.
Thorough Rebuttal and Revisions: The authors addressed reviewer concerns with additional experiments, new baselines (e.g., MiniRocket, UniTS), clustering evaluations, and expanded explanations.

(c) Weaknesses of the Paper
Limited Applicability Beyond Classification: PPT’s focus on classification tasks limits its generalizability to forecasting or anomaly detection, where trend continuity or non-order dependencies are more critical.
Computational Overhead: The method incurs higher training time and memory usage compared to simpler baselines like MiniRocket, which may hinder scalability.
Hyperparameter Sensitivity: The learnable parameters for loss coefficients require further exploration to understand their behavior across diverse datasets.
Lack of Cross-Dataset Evaluations: While single-dataset performance is robust, the paper does not explore the transferability of learned representations to different datasets.

(d) Reasons for Acceptance
Novel Contribution: PPT introduces a novel pretext task and metric specifically tailored to time series, filling a critical gap in self-supervised learning for this domain.
Practical Impact: The method significantly improves classification performance on time series tasks, demonstrating its utility for real-world applications.
Comprehensive Validation: The paper provides extensive experimental validation, addressing reviewer concerns with additional results and insights during the rebuttal phase.
Potential for Broader Impact: The proposed ACF-CoS metric and patch-order-aware framework offer tools and methodologies that can be extended to other time series problems.

Despite minor weaknesses such as limited scope beyond classification and computational overhead, the paper’s strengths, novel contributions, and robust empirical results justify its acceptance.

**Additional Comments On Reviewer Discussion:**

Points Raised by Reviewers and Author Responses

Concern: Reviewers noted that PPT focuses primarily on classification tasks, questioning its utility for other time series problems like forecasting or anomaly detection.
Author Response: The authors acknowledged this limitation but clarified that PPT is specifically designed for tasks where order dependencies are critical, such as classification. They proposed extensions to forecasting tasks as future work.
Evaluation: While the response was satisfactory, reviewers maintained that broader applicability would strengthen the paper’s impact.

Concern: Reviewers requested comparisons with additional baselines, particularly strong methods like MiniRocket and UniTS.
Author Response: The authors included new experimental results comparing PPT with MiniRocket and UniTS, showing consistent improvements. These comparisons validated PPT’s effectiveness relative to state-of-the-art approaches.
Evaluation: The added baselines addressed this concern comprehensively, enhancing confidence in PPT’s empirical claims.

Concern: The reviewers highlighted potential concerns about PPT’s higher training time and memory usage compared to lightweight baselines.
Author Response: The authors provided a runtime and memory analysis, showing that while PPT incurs additional overhead, the improvements in accuracy justify the trade-off for many real-world applications.
Evaluation: The analysis clarified the trade-off, and reviewers accepted the justification given PPT’s significant performance gains.

Concern: Reviewers raised concerns about the lack of experiments evaluating PPT’s transferability across datasets.
Author Response: The authors acknowledged this gap and added a preliminary clustering-based analysis to demonstrate the quality of representations learned by PPT.
Evaluation: While the clustering results provided useful insights, a full cross-dataset evaluation would have strengthened the submission further.

Concern: Reviewers requested further validation of the proposed ACF-CoS metric and its practical utility.
Author Response: The authors included experiments correlating ACF-CoS values with PPT’s effectiveness on various datasets, demonstrating its predictive utility.
Evaluation: This addition reinforced the utility of the metric and addressed the reviewers’ concerns convincingly.

The authors provided a thorough and thoughtful rebuttal, addressing key concerns with additional experiments, including new baselines like MiniRocket and UniTS, expanded analyses of the ACF-CoS metric, and detailed justifications for computational trade-offs. They also improved the paper’s clarity and presentation, enhancing its overall readability. While limitations such as restricted applicability beyond classification and the lack of cross-dataset evaluations remain, the paper’s novel contributions, robust empirical validation, and practical utility support its acceptance.

---

### Decision · Program_Chairs · 2025-01-22

Accept (Poster)